# Natural Language Actor-Critic:
## Scalable Off-Policy Learning in Language Space

## Abstract

Large language model (LLM) agents—LLMs that dynamically interact with an environment over long horizons—have become an increasingly important area of research, enabling automation in complex tasks involving tool-use, web browsing, and dialogue with people. In the absence of expert demonstrations, training LLM agents has relied on policy gradient methods that optimize LLM policies with respect to an (often sparse) reward function. However, in long-horizon tasks with sparse rewards, learning from trajectory-level rewards can be noisy, leading to training that is unstable and has high sample complexity. Furthermore, policy improvement hinges on discovering better actions through exploration, which can be difficult when actions lie in natural language space. In this paper, we propose *Natural Language Actor-Critic* (NLAC), a novel actor-critic algorithm that trains LLM policies using a generative LLM critic that produces natural language rather than scalar values. This approach leverages the inherent strengths of LLMs to provide a richer and more actionable training signal; particularly, in tasks with large, open-ended action spaces, natural language explanations for why an action is suboptimal can be immensely useful for LLM policies to reason how to improve their actions, without relying on random exploration. Furthermore, our approach can be trained off-policy without policy gradients, offering a more data-efficient and stable alternative to existing on-policy methods. We present results on a mixture of reasoning, web browsing, and tool-use with dialogue tasks, demonstrating that NLAC shows promise in outperforming existing training approaches and offers a more scalable and stable training paradigm for LLM agents.

## 1 Introduction

While LLMs excel at natural language tasks like question-answering (Pyatkin et al., 2022) and problem-solving (Hendrycks et al., 2021; Jimenez et al., 2024), which can be solved with a single response, LLM agent tasks require multi-turn interactions. Specifically, LLM agent tasks require the model to act within an environment, by taking actions sequentially and observing their results, ultimately to accomplish some long-term goal. Such tasks include autonomous reasoning (OpenAI, 2025), tool-use (Nakano et al., 2022), and dialogue with users (Hong et al., 2023; Yu et al., 2023). These tasks require agents to dynamically plan and intelligently respond to environmental stimuli, which base, pretrained LLMs struggle to do without additional training (Bachmann & Nagarajan, 2024). To train effective LLM agents, we will need algorithms that can fine-tune LLMs to pursue temporally extended goals in the context of multi-turn, long-horizon interactions.

Currently, LLM agents are trained with a variety of methods, often combining supervised fine-tuning (SFT) with reinforcement learning (RL) (Rafailov et al., 2023; Carta et al., 2023; Wang et al., 2025). For complex agentic tasks where labeled expert data is expensive to collect, such as ones involving interaction with real users, the prevailing training methods focus on policy optimization using algorithms such as Proximal Policy Optimization (PPO) (Schulman et al., 2017) or Group Relative Policy Optimization (GRPO) (Shao et al., 2024). The LLM agents are trained to generate environment actions accompanied by high-level reasoning to explain their decision-making (Yao et al., 2022; Wei et al., 2023). These methods are designed to teach an LLM to reason about the problem, plan over appropriate actions, and learn from the environment observations. However, prior attempts of using RL to train LLM agents present significant problems. First, these algorithms are notoriously data-inefficient because they are on-policy, meaning they require sampling new trajec-

tories from the current policy at every training step. Second, and perhaps more importantly, they rely on an often sparse, scalar reward as their only training signal, which can be a weak and unstable signal for learning robust, generalizable strategies over long-horizon tasks.

In this work, we propose a new actor-critic algorithm (Haarnoja et al., 2018) to train LLM agents, where a critic (which estimates the value of actions) is jointly learned with a policy, both using *off-policy* data. In contrast to traditional actor-critic to train LLM agents (Chebotar et al., 2023; Zhou et al., 2024b), we believe training a critic that outputs textual evaluations more effectively leverages the text-based reasoning capabilities of pretrained LLMs. Specifically, policy optimization using scalar values requires the policy to discover actions of high value through random exploration. If values were instead in natural language space, an LLM policy could understand how to improve its decision-making, reducing the reliance on random chance to uncover better actions.

Prior methods exist that train critics to generate language evaluations (Feng et al., 2025; Hong et al., 2025). Notably, Feng et al. (2025) propose Natural Language Reinforcement Learning (NLRL) as a framework for learning policies and critics in language space. Our work aims to address key limitations in NLRL to policy iteration in language space scalable to all LLM agent tasks. Specifically, NLRL relies on enumerating over both environment transitions, as well as potential actions, and aggregating them using in-context learning. We believe such training is impractical for tasks with complex dynamics or action spaces (such as dialogue), as enumerating over all possibilities and fitting them in-context is infeasible with limited time and memory. Our algorithm falls under the paradigm introduced by NLRL, but proposes novel objectives to train the critic and improve the policy that scales to learning general LLM agents.

In this paper, we propose Natural Language Actor-Critic (NLAC), a novel algorithm for training LLM agents, where a natural language critic is jointly trained with a policy, and its evaluations directly inform how to perform policy improvement. Theoretically, we are able to connect the learned representations of our critic to successor features (Barreto et al., 2017), allowing us to prove convergence to the optimal policy. Empirically, we evaluate our approach on a range of LLM agent tasks, ranging from reasoning, tool-use, and dialogue. Our empirical results demonstrate substantial improvement over prior approaches to learn LLM agents, showing our algorithm is an appealing alternative to prevailing on-policy training methods.

## 2 RELATED WORK

**LLM agents.** LLM agents can be used to tackle a variety of complex real-world tasks, including dialogue (Hong et al., 2023; Yu et al., 2023), tool-use (Nakano et al., 2022; Schick et al., 2023), and embodied decision-making (Wang et al., 2023). The primary challenge in the design of effective LLM agents is enabling LLMs, which traditionally excel at generating single-step responses, to interact sequentially with an environment to accomplish a long-term objective. ReAct prompting is a popular method to leverage chain-of-thought reasoning of LLMs for long-horizon planning, by instructing LLMs to explicitly articulate their high-level plans (Yao et al., 2022). More recent approaches have explored the capability of LLM agents to self-correct their initial attempts at planning using more sophisticated prompting techniques (Shinn et al., 2023; Madaan et al., 2023; Zhou et al., 2024a). For example, Reflexion prompting adds a step of self-reflection on top of ReAct to allow LLM agents to refine their initial reasoning after some environment feedback (Shinn et al., 2023). However, self-correction methods rely the ability to "backtrack," or undo previous actions, whereas we measure the capability of LLM agents with one chance to solve a task.

**Process reward models.** One of the primary challenges in learning LLM agents is the reliance on a single, sparse reward for long-horizon interactions. This makes credit assignment, or distinguishing between good and bad actions in a long rollout, difficult. Process reward models (PRMs) aim to address this, particularly by providing action-level feedback using either human annotations (Lightman et al., 2023), or an estimated value function in the absence of human intervention (Wang et al., 2024; Setlur et al., 2025). Our learned natural language critic can be considered an instance of an PRM, but unlike traditional PRMs that provide scalar feedback over actions, our critic outputs feedback in language space. We believe such feedback is more useful for LLM policies that can understand and articulate their decisions in natural language.

**Reinforcement learning for LLM agents.** More recently, multiple works have attempted to explicitly fine-tune LLMs as agents using RL (Carta et al., 2024; Zhou et al., 2024b). The primary way this was done was naively adapting traditional RL fine-tuning used to align LLM responses to multi-turn

tasks with environment interaction (Stiennon et al., 2020; Ouyang et al., 2022; Ramamurthy et al., 2023). These methods used PPO (Schulman et al., 2017) to finetune LLMs using the environment reward. However, traditional policy optimization for long-horizon tasks exacerbates the instabilities of RL training, particularly due to reliance on exploration and proper credit assignment. In this work, we hypothesize that training in natural language over scalar space improves stability and sample efficiency, particularly in better leveraging the capabilities of LLMs to understand and articulate thoughts in natural language. The closest work to ours that does this is NLRL (Feng et al., 2025), which also proposes learning value functions that output text. However, in NLRL, these values are obtained via repeated sampling of on-policy trajectories and aggregating them in-context. In addition, policy improvement is achieved by enumerating over possible actions and their evaluations. We believe such enumeration and aggregation in-context is intractable for tasks with complex dynamics and large action spaces. Our method circumvents these drawbacks by training the critic to probabilistically generate textual rollouts via a novel language Bellman backup, and treating policy improvement as iterative refinement.

## 3 PRELIMINARIES

**Markov decision processes.** We adopt the formalism of a Markov decision process (MDP) given by $M = (\mathcal{S}, \mathcal{A}, P, r, \rho, \gamma)$, where $\mathcal{S}$ is the state space, $\mathcal{A}$ is the action space, $P$ is the transition function, $r$ is the reward function, $\rho$ is the initial state distribution, and $\gamma$ is the discount factor. When action $a \in \mathcal{A}$ is executed at state $s \in \mathcal{S}$, the next state is sampled $s' \sim P(\cdot|s, a)$, and the agent receives reward $r$ with mean $r(s, a) \in \mathbb{R}$.

**LLM agents in MDPs.** Tasks considered by LLM agents can be defined under the MDP formalism as follows. Here, the state and action space are finite-length sequences of tokens in vocabulary $\mathcal{V}$, or $\mathcal{S}, \mathcal{A} \subseteq \mathcal{V}^*$, where $\mathcal{V}^*$ denotes all finite sequences comprised of tokens in vocabulary $\mathcal{V}$. We also define the space of environment observations $\mathcal{O} \subset \mathcal{V}^*$; those could consist of results of API calls in tool-use applications, or responses by other interlocutors in dialogue. The agent corresponds to a policy $\pi$ that starts by observing a task description along with any initial observations $s_1 = (q, o_0)$. At timestep $t$, the agent *state* $s_t$ of the MDP consists of the history of interaction thus far $s_t = (q, a_1, o_1, \ldots, a_{t-1}, o_t)$ consisting of agent actions and environment observations.

**ReAct prompting.** LLM agents are commonly implemented using ReAct prompting to better leverage the base reasoning capabilities of LLMs (Yao et al., 2022). ReAct prompting instructs LLM agents to output actions $a_t \sim \pi(\cdot|s_t)$ that are actually composite, consisting of a *thought* $a_t^{\mathrm{tht}}$ where the agent performs a reasoning step, followed by the actual environment action $a_t^{\mathrm{env}}$. For example, in dialogue, the thought could be the high-level strategy or plan the agent aims to execute, whereas the environment action is the actual utterance by the agent. Then, the transition function appends to $s_t$ the environment action $a_t^{\mathrm{env}}$ as well as any new observations by the environment $o_{t+1}$, to form the next state $s_{t+1}$. Note that the thought does not affect the transition dynamics, namely $P(\cdot|s_t, a_t) = P(\cdot|s_t, a_t^{\mathrm{env}})$.

**Reinforcement learning.** The objective of RL is to find a policy $\pi$ that maximizes the expected discounted return $J(\pi) = \mathbb{E}_{\tau \sim p^\pi} \left[ \sum_{t=0}^{T-1} \gamma^t r(s_t, a_t) \right]$ in an MDP, where $\tau = (s_0, a_0, s_1, a_1, \ldots, s_T)$ and $p^\pi(\tau) = \rho(s_0) \prod_{t=0}^{T-1} \pi(a_t|s_t) P(s_{t+1}|s_t, a_t)$. Standard policy gradient approaches directly train policy $\pi$ using the gradient of $\nabla_\pi J(\pi)$, while more sophisticated algorithms such as PPO and GRPO additionally clip the updates to improve stability (Schulman et al., 2017; Shao et al., 2024). Actor-critic algorithms additionally learn a state-action value function, or Q-function, defined as $Q^\pi(s_t, a_t) = \mathbb{E}_{(s,a)_{t+1:\infty} \sim p^\pi} \left[ \sum_{t'=t}^{T-1} \gamma^{t'-t} r(s_{t'}, a_{t'}) \right]$. Such Q-functions are learned by regressing to their Bellman backup:

$$\mathcal{B}Q^\pi(s_t, a_t) = r(s_t, a_t) + \mathbb{E}_{s_{t+1}, a_{t+1} \sim P^\pi} \left[ Q^\pi(s_{t+1}, a_{t+1}) \right],$$

where $P^\pi(s', a'|s, a) = P(s'|s, a)\pi(a'|s')$. Then, an improved policy $\pi'$ can be derived using the Q-function via greedy or maximum-entropy optimization $\pi'(a_t|s_t) \propto \exp(Q^\pi(s_t, a_t))$.

## 4 NATURAL LANGUAGE ACTOR-CRITIC

In this section, we present Natural Language Actor-Critic (NLAC), our new method for training LLM agents that adopts the actor-critic paradigm. Unlike traditional methods that rely on simple policy gradients, NLAC leverages a *natural language critic* that outputs textual critiques of actions

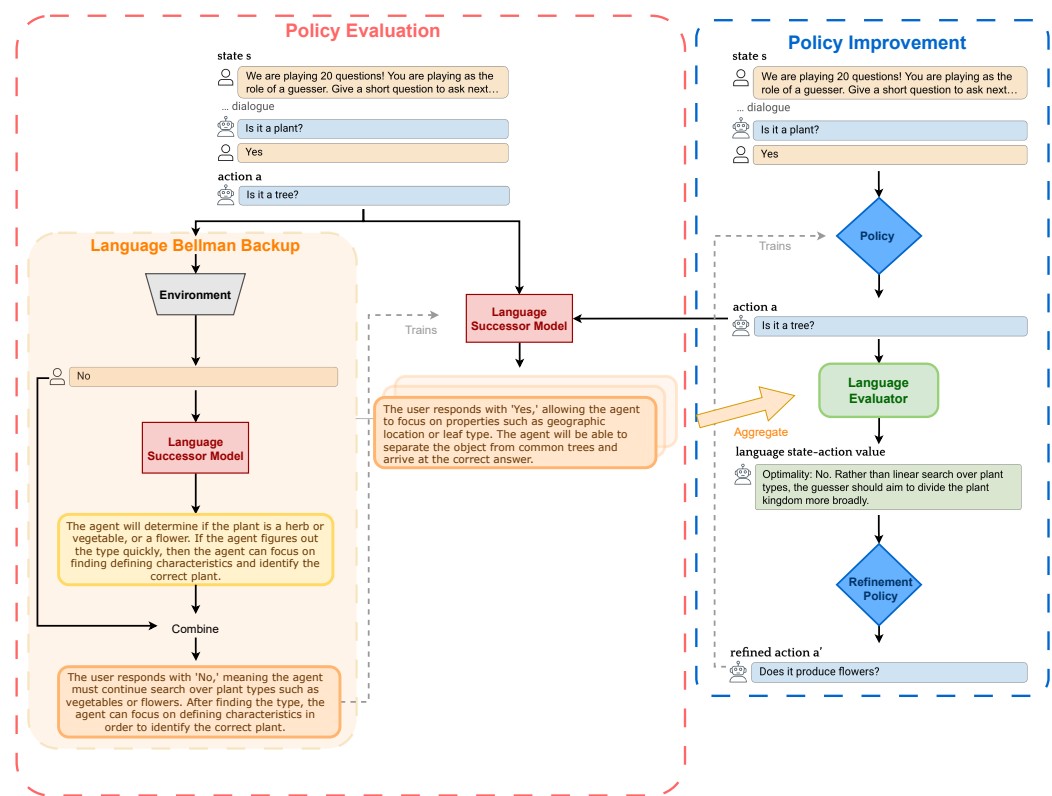

Figure 1: Overview of NLAC. During *policy evaluation*, the critic is trained using a language Bellman backup that operates in textual space. During *policy improvement*, the policy is distilled from a refinement policy.

to provide a rich, interpretable, and more stable training signal. Our framework is inspired by classical actor-critic methods where each step consists of (1) policy evaluation, where a critic is trained to assess actions by a policy, and (2) policy improvement, where the policy is updated using evaluations by the critic, but is adapted to leverage the implicit reasoning capabilities of LLMs over text space. In our approach, both the LLM policy and the natural language critic are instantiated by the same underlying LLM, with their distinct functionalities realized through different prompts. We go over both components in detail below.

## 4.1 POLICY EVALUATION

In traditional actor-critic approaches, a critic is trained to estimate scalar state-action values, or Q-values, typically denoted as $Q^\pi(s, a) \in \mathbb{R}$, which represents the expected return by policy $\pi$ from state $s$ after taking action $a$. While learning such Q-values can be similarly done with LLM critics, LLMs are better suited to process and generate natural language over scalars. Therefore, we believe evaluation that is in natural language space leverages prior text-based reasoning capabilities of LLMs, and thus will largely improve sample efficiency. Hence, our natural language critic is an LLM that generates textual critiques, denoted as $Q_L^\pi(s, a) \in \mathcal{V}^*$, that not only comments on how good an action is, but also explains why.

**Predicting the future using language.** The key addition that is not captured by scalar Q-values is an explanation of why a particular action is optimal or not. As we will discuss later, this information is valuable for LLM policies to understand how to refine their actions during policy improvement, avoiding the reliance on random exploration to discover better actions. We believe that the key for a critic to derive these explanations is the prediction and analysis of future outcomes. In order to do so, we must train our natural language critic to additionally act as a successor function, defined as follows:

**Definition 4.1.** *A **language successor model** $M^\pi$ for policy $\pi$ takes a state $s_t$ and action $a_t$ as input, and probabilistically generates a textual description of rollout $(s, a)_{t+1:T}$, or what will happen to*

*policy $\pi$ in the future, and reward $r(s_T)$. We denote by $M^\pi(\cdot \mid s_t, a_t)$ the distribution from which such descriptions are sampled.*

Our language successor model shares similarities with successor features (Barreto et al., 2017) in that both can predict a distribution over future rollouts, and—as we show later—be trained using temporal difference learning. The main difference lies in that traditional successor features are used to compute Q-values via a linear product, whereas ours is used to generate state-action values in natural language via output by an LLM.

To train our language successor model, we draw inspiration from distributional value learning (Bellemare et al., 2017), which introduces a distributional Bellman backup to train a distribution over returns rather than just their scalar expectation. Notably, the distributional Bellman backup used one-step samples of the future and thus could be computed off-policy. Similarly, we propose a *language Bellman backup* $\mathcal{B}_L$ that bears some semblance to the distributional Bellman backup, but makes key adaptations to account for samples that are textual descriptions of rollouts rather than scalar returns.

**Definition 4.2.** *A **language Bellman backup** $\mathcal{B}_L$ takes a language successor model $M^\pi$, along with state $s_t$ and action $a_t$ as input, and computes distribution $\mathcal{B}_L M^\pi(\cdot|s_t, a_t)$ such that the probability of description $d_t \in \mathcal{V}^*$ is given by:*

$$\mathcal{B}_L M^\pi(d_t \mid s_t, a_t) = \Pr\left[d_t = B(r(s_t, a_t), s_{t+1}, a_{t+1}, d_{t+1}) \mid s_{t+1}, a_{t+1}, d_{t+1}\right], \quad (1)$$

$$s_{t+1}, a_{t+1} \sim P^\pi(\cdot \mid s_t, a_t), \; d_{t+1} \sim M^\pi(\cdot \mid s_{t+1}, a_{t+1}),$$

*where $B$ is a function that combines immediate next state and action $s_{t+1}, a_{t+1}$ with description $d_{t+1}$ of rollout $(s, a)_{t+2:T}$ into one description of the concatenated rollout $(s, a)_{t+1:T}$.*

Beyond simple concatenation, the $B$ function "discounts" the future rollout description from $M^\pi$ in the concatenated rollout so the immediate next state is given more emphasis in the description.

Then, we can train our language successor model $M^\pi$ by minimizing the divergence between distributions $M^\pi(\cdot|s_t, a_t)$ and target distributions created by the language Bellman backup:

$$M^\pi = \arg\min_M \mathbb{E}_{(s_t, a_t s_{t+1}) \sim \mathcal{D}}\left[D_f\left(M(\cdot \mid s_t, a_t) \,\|\, \mathcal{B}_L M(\cdot \mid s_t, a_t)\right)\right]. \quad (2)$$

Note that our training objective is an instance of temporal-difference learning and thus does not require on-policy Monte Carlo trajectories.

**Generating critiques.** Finally, the natural language critic should analyze all possible futures in order to evaluate how good an action is in expectation, then explain its reasoning by referencing possible future outcomes. To perform this evaluation, we define the following:

**Definition 4.3.** *A **language evaluator** $E$ takes as input state $s_t$ and action $a_t$, along with a sequence of descriptions of possible rollouts $(s, a)_{t+1:T}$ and their rewards $r(s_T)$, and outputs a textual critique that comments on whether $a_t$ was optimal, with justification using possible future outcomes.*

Then, we can approximate natural language value $Q_L^\pi(s_t, a_t)$ as:

$$Q_L^\pi(s_t, a_t) \approx E(s_t, a_t, d_t^{(1)}, \dots, d_t^{(k)}), \quad d_t^{(i)} \sim M^\pi(\cdot \mid s_t, a_t), \; \forall i \in [k]. \quad (3)$$

Note that $E$ essentially aggregates and summarizes multiple descriptions of different rollouts that are all fit in-context, which LLMs have demonstrated a priori efficacy in without additional training (Feng et al., 2025). This means that the only training required to perform evaluation of policy $\pi$ in language space is learning the language successor model $M^\pi$ (see Figure 1 for illustration).

### 4.2 POLICY IMPROVEMENT

Thus far, we showed how to train the natural language critic to evaluate a fixed policy $\pi$. We now show how an improved policy can be learned using textual critiques $Q_L^\pi(s, a)$ obtained by a critic using Equation 3. Naturally, such policy is a greedy policy where $a \sim \pi(\cdot|s)$ satisfies $a = \arg\max_{a'} Q_L^\pi(s, a')$. Note that we assume the following:

**Assumption 4.1.** *For any policy $\pi$, the set $\{Q_L^\pi(s, a')\}_{a' \in \mathcal{A}}$ computed using Equation 3 for any state $s$ forms a totally-ordered set with binary relation $\geq$.*

We believe that this is not a strong assumption, as each critique $Q_L^\pi(s, a)$ can be mapped to a scalar that quantifies its sentiment, which can be used to compare with other critiques. Then, $Q_L^\pi(s, a') \geq Q_L^\pi(s, a)$ if the underlying sentiment of the text in $Q_L^\pi(s, a')$ is more positive.

However, computing the greedy policy is intractable for LLM agent tasks, where the action spaces $\mathcal{A} \subseteq \mathcal{V}^*$ are combinatorial in the token vocabulary, making it impossible to enumerate all possible actions to find the optimal one. While prior works have proposed sampling a subset of actions and reweighting (Li et al., 2024), we find empirically that for tractable sample sizes, this approach does not sufficiently explore the space of possible actions.

Our approach sidesteps this issue by leveraging the descriptive power of the natural language values using a self-refinement paradigm. Our insight is that the natural language value $Q_L^\pi(s, a)$ not only comments on how good an action is, but also contains intuition on how a suboptimal action can be improved. Hence, a policy that is an LLM with strong base reasoning capabilities can process this evaluation and understand how to *refine* its initial action.

To this end, we define a *refinement policy* $\pi^r$ that takes an action $a_t \sim \pi(\cdot|s_t)$ by the base policy, and generates a refined action $a_t^r \sim \pi^r(\cdot|s_t, a_t, Q_L^\pi(s_t, a_t))$ that is better according to the natural language critic, i.e., $Q_L^\pi(s_t, a_t^r) \geq Q_L^\pi(s_t, a_t)$. As with the policy and critic, the refinement policy can use the same underlying LLM but with a different prompt. Note that refinement can also be performed iteratively by maintaining and appending to a history of all previous action attempts and their evaluations

$$a_t^r \sim \pi^r(\cdot \mid s_t, a_t^1, Q_L^\pi(s, a_t^1), \ldots a_t^m, Q_L^\pi(s_t, a_t^m)),$$

where we can control for a parameter $m$ that denotes number of rounds of refinement. As $m \to \infty$, we expect the refined action $a_t^r$ to be the greedily optimal one $a_t^r = \arg\max_a Q_L^\pi(s_t, a)$.

Finally, we propose a policy improvement objective from $\pi$ to $\pi'$ that projects the refinement policy back to the base policy, similar to the policy updates in SAC (Haarnoja et al., 2018). However, rather than parameterizing a target policy using the learned values, which requires enumeration over actions and is intractable in our setting, we let the target policy be the refinement policy:

$$\pi' = \arg\max_\pi \mathbb{E}_{s_t \sim \mathcal{D}} \left[ D_f \left( \pi(\cdot \mid s_t) \,\|\, \pi^r(\cdot \mid s_t, a_t^1, \ldots, Q_L^\pi(s, a_t^m))) \right) \right] . \tag{4}$$

In practice, we found that a single round of refinement $m = 1$ was sufficient. Again, this objective does not require any on-policy rollouts, and can therefore be trained off-policy. This refinement is visualized in Figure 1.

## 5 THEORETICAL ANALYSIS

The goal of this section is to show that policy iteration using our proposed NLAC method converges to the optimal policy. Due to space, we only state our main results and defer full proofs to Appendix A. For traditional actor-critic algorithms, this involves showing (1) convergence of learned Q-values via the Bellman backup, and (2) monotonic improvement of the trained policy. However, such analysis does not apply because the Q-values we learn are textual rather than scalar.

Instead of analyzing the textual values, we consider the underlying representations that the LLM decodes in order to generate such values. First, we define $\phi(s) \in \mathbb{R}^d$ as features extracted from any state $s$. We assume the following:

**Assumption 5.1.** *The expected reward $r(s, a)$ for any state $s$ and action $a$ can be linearly represented by the features as $r(s, a) = \phi(s) \cdot w$ for some fixed $w \in \mathbb{R}^d$.*

Next, we define representations $\Phi^\pi(s, a) \in \mathbb{R}^d$ such that we can write the output of our language successor model as $M^\pi(\cdot|s, a) = f_M(\Phi^\pi(s, a))$, and similarly $Q_L^\pi(s, a) = f_Q(\Phi^\pi(s, a))$, for some functions $f_M, f_Q$ denoting decoding by the LLM. We make the following assumption about the effect of our proposed language Bellman backup on such representations:

**Assumption 5.2.** *For any state $s$ and action $a$ , the language Bellman backup satisfies*

$$\mathcal{B}_L M^\pi(\cdot|s, a) = f_M \left( \phi(s) + \gamma \mathbb{E}_{s', a' \sim P^\pi} \Phi^\pi(s', a') \right) .$$

While this may initially seem like a strong assumption, note that our language Bellman backup is already instructed to combine the immediate observation with the future description in language space; the assumption only states that the combination also corresponds to a discounted sum in the representation space. Using the above two assumptions, our first main result is the following:

**Theorem 5.1.** *Consider policy evaluation via Equation 2 and let $Q_L^\pi$ be the natural language critic at convergence. For any state $s$ and action $a$, there exists monotonic mapping $g$ such that $Q^\pi(s, a) = g(Q_L^\pi(s, a))$, where $Q^\pi$ denotes the true scalar Q-function.*

Our main result makes a precise connection between our learned critic and the true Q-function. The crux of our proof involves showing that our training objective results in a fixed point where $\Phi^\pi(s, a)$ are *successor features* of the underlying MDP (Barreto et al., 2017).

During policy improvement, we update the policy towards a refinement policy $\pi^r$. By definition, the refinement policy is an oracle that for any state $s$ and action $a$, generates $a^r$ such that $Q_L(s, a^r) \geq Q_L(s, a)$. Combining this with the result of Theorem 5.1, we arrive at our second main result:

**Theorem 5.2.** *Repeated application of policy evaluation via Equation 2 and policy improvement via Equation 4 from a policy $\pi_0$ converges to policy $\pi^*$ such that $Q^{\pi^*}(s, a) \geq Q^\pi(s, a)$ for any state $s$ and action $a$, and other policy $\pi$.*

Hence, we are able to show that under the aforementioned assumptions, our approach NLAC can provably find the optimal policy for an underlying MDP. Next, we show how to approximately implement NLAC in a scalable and practical algorithm.

## 6 PRACTICAL IMPLEMENTATION

In this section, we describe how both the critic and policy are trained in practice. We defer specific details such as exact prompts used to Appendix B. Though our method involves many different components such as a language successor model and evaluator, we can leverage the general capabilities of LLMs to reason over and generate language to reuse one model to implement all the described components. Hence, our algorithm only involves training one LLM $\mathcal{M}$ with parameter $\theta$. For a prompt $p \in \mathcal{V}^*$, we denote by $\mathcal{M}_\theta(p)$ the distribution over responses by the LLM.

### 6.1 TRAINING COMPONENTS

**Policy.** Many prior works have parameterized policies as LLMs. One of the greatest advantages of doing so is the ability to leverage the strong reasoning capabilities of LLMs from chain-of-thought prompting Wei et al. (2023); Yao et al. (2022). By choosing a proper prompt $p_{\text{react}}$, an LLM policy can be instructed to describe their underlying thoughts for choosing a particular action in addition to generating the action itself $a_t \sim \mathcal{M}_\theta(p_{\text{react}}(s_t))$.

**Language successor model.** LLMs have demonstrated efficacy at predicting realistic future rollouts in a variety of environments (Lin et al., 2024). These futures are generated by simply processing the state-action in a prediction prompt $p_{\text{pred}}$ that also instructs the LLM to summarize rollouts into concise textual descriptions, then sampling from the LLM output $M_\theta(\cdot \mid s_t, a_t) = \mathcal{M}_\theta(p_{\text{pred}}(s_t, a_t))$.

**Language Bellman backup.** The backup $\mathcal{B}_L$ also outputs a distribution over descriptions of rollouts, but uses one-step samples of next state along with a "bootstrapped" description of rollout generated by $M_\theta$. We give the LLM instruction $p_{\text{tpred}}$ to predict a "target" future by combining the immediate next state with the bootstrapped future description into one description, discounting the future description as necessary by placing more emphasis on the immediate next state.

$$\mathcal{B}_L M_\theta(\cdot \mid s_t, a_t) = \mathcal{M}_\theta(p_{\text{tpred}}(r_t, s_{t+1}, d_{t+1})), \quad d_{t+1} \sim \mathcal{M}_\theta(p_{\text{pred}}(s_{t+1})).$$

Note that we do not explicitly sample $a_{t+1}$ from the policy, but implicitly via the language successor model that is conditioned on the policy.

**Language evaluator.** The evaluations by $E$, which ultimately become the outputs of the natural language critic that estimate $Q_L^\theta(s_t, a_t)$ can similarly be derived by fitting multiple generated futures $d_t^{(1)}, \ldots d_t^{(k)}$ in-context within an evaluation prompt $p_{\text{eval}}$ that asks the LLM to aggregate the futures and summarize into an overall description of how good the action is, as

$$Q_L^\theta(s_t, a_t) = E_\theta(s_t, a_t, d_t^{(1)}, \ldots, d_t^{(k)}) \sim \mathcal{M}_\theta(p_{\text{eval}}(d_t^{(1)}, \ldots, d_t^{(k)})).$$

**Refinement policy.** Finally, the refinement policy $\pi^r$ can also be obtained by an LLM instructed to refine its latest action given an evaluation similar to prior self-refinement approaches (Madaan et al., 2023). The refined action is obtained via prompt $p_{\text{refine}}$ as $a_t^r \sim \mathcal{M}_\theta(p_{\text{refine}}(s_t, a_t^1, \ldots, Q_L^\theta(s_t, a_t^m)))$.

### 6.2 TRAINING ALGORITHM

Formally, the parameters $\theta$ are trained using two objectives for policy evaluation and improvement. For policy evaluation, for a transition $(s_t, a_t, s_{t+1})$, the natural language critic is trained using cross entropy component of the objective:

$$\mathcal{L}_1(s_t, a_t, r_t, s_{t+1}) = D_{\mathsf{KL}}\left(\mathcal{B}_L M_{\bar{\theta}}(\cdot \mid s_t, a_t) \,\|\, M_\theta(\cdot \mid s_t, a_t)\right), \tag{5}$$

where $\bar{\theta}$ are reference parameters that are an exponentially moving average of the trained parameters, in order to prevent generative collapse (Shumailov et al., 2024). We choose the reverse direction of KL-divergence to capture the full diversity over possible futures. Then, for policy improvement, we train the policy on the log-likelihood loss:

$$\mathcal{L}_2(s_t, k) = -\log \pi_\theta(a_t^r \mid s_t), \quad a_t \sim \pi_\theta(\cdot \mid s_t), \ a_t^r \sim \pi_\theta^r(\cdot \mid s_t, a_t, Q_L^\theta(s_t, a_t)). \ (6)$$

This objective can be interpreted as distillation, but using generations by the refinement policy as the teacher policy. Note that the loss depends on $k$ via $Q_L^\theta(s_t, a_t)$ given by Equation 3. Note that by default, our refinement policy relies on the base reasoning capabilities of the pretrained LLM. In Appendix B.5, we show results when the refinement policy is explicitly trained.

We show pseudocode for NLAC in Algorithm 1. In practice, we found it helpful to implement $\mathcal{D}$ as a prioritized replay buffer weighted by $\mathcal{L}_1(s_t, a_t, s_{t+1})$ with sampling parameter $\alpha$ (Schaul et al., 2016). This is because in many tasks, though a base LLM policy may achieve low reward in a large proportion of rollouts, many actions in these unsuccessful rollouts are still optimal. Therefore, to improve learning efficiency, we prioritize training on samples where the agent is likely to take a suboptimal action, using critic loss as a proxy for the likelihood.

---

**Algorithm 1** Natural Language Actor-Critic (NLAC)

---

1: Initialize $\theta, \bar{\theta}$ from pretrained model.
2: **for** each iteration **do**
3:    **for** each environment step **do**
4:       Sample $a_t \sim \pi_\theta(\cdot \mid s_t), s_{t+1} \sim P(\cdot \mid s_t, a_t)$
5:       Add to replay buffer $\mathcal{D} \leftarrow \mathcal{D} \cup \{(s_t, a_t, r_t, s_{t+1})\}$
6:    **end for**
7:    **for** each training sample **do**
8:       $\theta \leftarrow \theta - \lambda_1 \nabla_\theta \mathcal{L}_1(s_t, a_t, r_t, s_{t+1})$
9:       $\theta \leftarrow \theta - \lambda_2 \nabla_\theta \mathcal{L}_2(s_t, k)$
10:      $\bar{\theta} \leftarrow \tau\theta + (1 - \tau)\bar{\theta}$
11:    **end for**
12: **end for**

---

Like other methods that utilize pretrained LLMs, our method is susceptible to catastrophic forgetting. We were able to avoid this in our experiments by training in low-data regimes. However, we discuss effective methods for mitigating catastrophic forgetting in Appendix B.6.

# 7 EXPERIMENTS

To demonstrate the effectiveness of NLAC, we evaluate our method on a variety of LLM agent tasks: mathematical reasoning (Hendrycks et al., 2021), strategic dialogue (Pu, 2023), and customer service using mixed dialogue and tool-use (Yao et al., 2024). Though mathematical reasoning does not involve interaction with an environment, it is currently the most popular benchmark to evaluate different RL fine-tuning algorithms.

## 7.1 TASK DESCRIPTIONS

**Mathematical reasoning.** We evaluate on mathematical problem-solving using the MATH dataset (Hendrycks et al., 2021), which consists of different competition math problems of varying level of difficulty. A score of 1 is achieved if the agent solves the problem and outputs an answer that is correct and properly formatted. We evaluate on a subset of 500 problems from the test dataset of the highest difficulty level, which we call MATH500-Hard. The remaining 12,000 problems are used as the training set for RL fine-tuning.

**Dialogue game.** We use the popular game of 20 Questions (20Q) as a representative strategic dialogue task, where the LLM agent acts as the guesser to uncover the hidden word by an oracle. 20Q was chosen because it was non-adversarial (so we can evaluate against a fixed LLM as the oracle), and requires the LLM agent to generate a cohesive sequence of actions over multiple steps. Though many implementations exist (Srivastava et al., 2023; Abdulhai et al., 2023), we follow the one by Pu (2023) where the set of hidden words can be any in a set of 1,823 objects from the THINGS dataset (Hebart et al., 2019). A reward of 1 is achieved if the guesser correctly identifies the hidden object within 20 turns, or questions, where correctness if determined by using the oracle LLM as a judge. We use GPT4.1 (OpenAI et al., 2024) as the oracle. We construct a training set of 1,000 objects and a test set of 500 different objects through random sampling.

**Customer service.** We consider $\tau$-bench as a representative LLM agent task that requires a mixture of dialogue and tool-use to solve (Yao et al., 2024). The LLM agent must act as a customer service

| Paradigm | Method | MATH500-Hard | 20Q | $\tau$-Bench | |
| | | Accuracy | Winrate | Retail | Airline |
|---|---|---|---|---|---|
| Prompting GPT4.1 | ReAct (Yao et al., 2022) | **95.1** | 30.2 | 0.44 | 0.32 |
| Fine-tuning Qwen2.5-7B-Instruct | RFT | 52.5 | 12.6 | 0.21 | 0.13 |
| | PPO (Schulman et al., 2017) | 52.3 | 17.2 | 0.28 | 0.14 |
| | GRPO (Shao et al., 2024) | 49.8 | 18.4 | 0.24 | 0.11 |
| | SAC (ablation) | 48.2 | 9.8 | 0.18 | 0.11 |
| | NLRL (Feng et al., 2025) | 62.4 | 25.8 | 0.25 | 0.16 |
| | NLAC (ours) | 60.2 | 26.0 | 0.42 | 0.22 |
| Fine-tuning QwQ-32B | RFT | 72.5 | 22.0 | 0.35 | 0.29 |
| | PPO (Schulman et al., 2017) | 71.4 | 24.0 | 0.47 | 0.41 |
| | GRPO (Shao et al., 2024) | 70.8 | 25.6 | 0.48 | 0.39 |
| | SAC (ablation) | 64.7 | 13.2 | 0.31 | 0.21 |
| | NLRL (Feng et al., 2025) | 73.5 | 30.8 | 0.44 | 0.31 |
| | NLAC (ours) | 72.7 | **32.1** | **0.59** | **0.45** |

Table 1: Performance on evaluation set of each benchmark. To make comparisons fair, each of the fine-tuning methods train for $30,720$ gradient steps, and we average performance across three independent runs. Across the board, NLAC outperforms other RL methods for both small- and medium-sized LLMs, and even beats frontier models on long-horizon tasks. Note that because mathematical reasoning is a single-step task, we have to adapt the language generative model to only predict reward.

representative in various scenarios such as modifying items in an user's order, and follow a rigid set of policy guidelines. At every step, the LLM agent can either communicate with the user, or make an API call that interacts with a backend database. At the end, the agent receives a score of 1 if the database entries match ground-truth values, and the agent did not violate any policy guidelines via their actions. Users are simulated using a GPT4.1 (OpenAI et al., 2024) model prompted with both an initial request (such as modifying or cancelling an order) as well an identity that can be verified using the database. There are two categories of scenarios: (1) in retail, the LLM agent must modify pending orders of items, return or exchange delivered orders, or update user information, and (2) in airline, the LLM agent must book, modify, or cancel flight reservations. To test generalization, we compile a training dataset of $2,500$ user scenarios in the retail category, and evaluate on a test set of 500 different retail scenarios, as well as 500 airline scenarios. Note that none of the methods are trained on any airline scenarios.

## 7.2 RESULTS

We compare NLAC with $k = 1$ and $m = 1$ against both prompting and fine-tuning baselines. We found those settings of hyperparameters was sufficient to achieve good performance, though more stochastic environments may warrant higher $k$. For baselines that involve fine-tuning, we consider two LLMs: Qwen2.5-7B-Instruct (Yang et al., 2024), and QwQ-32B (Team, 2025), which is also trained on reasoning traces. We choose these two LLMs to measure the effect of increasing size and pre-training on reasoning traces on the performance of the RL methods. Our baselines can be categorized into the following (training details can be found in Appendix B.4):

**Prompting.** We perform ReAct prompting (Yao et al., 2022) of a state-of-the-art frontier model GPT4.1 (OpenAI et al., 2024). Because such models do not expose weights for RL fine-tuning, we rely on the zero-shot capabilities of the LLM without any additional training on the tasks.

**Rejection fine-tuning.** We perform rejection fine-tuning (RFT) where at every iteration, the base LLM policy collects a set of on-policy rollouts. We append only the successful rollouts to a buffer, then train the LLM using SFT on samples from the buffer.

**RL fine-tuning.** The most standard way to perform RL fine-tuning is to train the LLM to optimize score using a policy gradient algorithm on on-policy rollouts. We consider both PPO (Schulman et al., 2017) and GRPO (Shao et al., 2024) as the algorithm, the difference being that PPO additionally learns a token-level value function on Monte-Carlo rollouts as a baseline to stabilize reward, whereas GRPO computes the average reward across 8 different rollouts. We found that increasing the number of rollouts for GRPO only harmed performance.

**Ablations.** We consider an ablation of our approach that is soft actor-critic (SAC) training. Instead of performing policy iteration in language space, SAC simply learns scalar values via a token-level Q-function using traditional Bellman backups, and then performs policy extraction by fitting the policy to the maximum-entropy policy using the learned Q-function (Haarnoja et al., 2018). Finally, we compare against NLRL (Feng et al., 2025); during training, instead of using the language Bellman backup to train a successor function and refinement to train the policy, NLRL aggregates environment transitions in-context with predicted future value, then enumerates multiple actions and their values, respectively. To keep NLRL tractable, as the state and action space are prohibitively large, we limit to 8 transitions and actions.

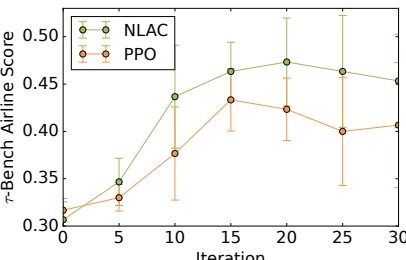

Figure 2: Learning curves for NLAC and PPO across three independent runs. NLAC converges in fewer samples.

The results of our evaluation are presented in Table 1. We see that for both LLM models, NLAC outperforms other fine-tuning approaches, and even prompting of a significantly stronger LLM, particularly on long-horizon tasks. The only task where NLAC matches other methods is mathematical reasoning, which is a single-step task, in which NLAC reduces to performing self-refinement using a generative reward model (Madaan et al., 2023); this is because in single-step tasks, our natural language critic is only trained to predict reward. Meanwhile, on tasks requiring multi-step interaction, which our method is tailored for, NLAC greatly outperforms all baselines, achieving a 30% improvement in 20Q and $\tau$-retail over standard RL fine-tuning. Furthermore, as shown in Figure 2, NLAC also requires fewer gradient steps to achieve maximum performance, illustrating the effectiveness of leveraging text-based reasoning. Surprisingly, ablation SAC performed worst; this can be attributed to the fact that token-level Q-values are difficult to estimate precisely, so directly fitting the policy to these Q-values can hurt performance. NLRL performed comparably to NLAC on math and 20Q, though it is important to note that NLRL effectively sees $8\times$ more samples during training. However, NLRL performed noticeably worse on $\tau$-bench. Qualitatively, we found that values learned by NLRL were almost always positive, which made them not helpful for policy improvement; this could be due to a combination of poor modeling of environment dynamics due to limited in-context samples, and the implicit bias of instruction fine-tuned models. Overall, this serves as evidence that NLAC is more suitable to complex tasks.

We demonstrate qualitative examples of how NLAC improves the base LLM policy in Figure 3 and Figure 4 for 20Q and $\tau$-bench, respectively (at Appendix C due to space). In 20Q, base LLM agents often resort to linear search over some specific characteristic of an object, when it is likely more optimal to further explore over other discriminators. In the example, the critique by our natural language critic explicitly corrects this linear strategy when it occurs. Meanwhile, in $\tau$-bench, one of the most common failure modes is partial resolution of complex requests, especially when the agent must also follow complicated dynamics and rules. In the example, the agent is told that the user wants to make "a couple of exchanges," but according to policy guidelines, modifications to the database can only be done via one tool-call per rollout. Therefore, the agent should not make a tool-call to exchange the first item, but instead collect all items to be exchanged into a single call in the future. This kind of error would be difficult to correct with just a scalar reward as feedback. However, the critique by our natural language critic identifies exactly which policy guideline would be violated, allowing for the LLM agent to easily understand and correct the error.

# 8 DISCUSSION

In this paper, we propose NLAC, a new actor-critic algorithm for training LLM agents under the paradigm of learning values in natural language space. In our work, the natural language values not only comment on the optimality of an action, but also articulate why by predicting and analyzing future outcomes. The key innovation we propose to enable this is a novel *language Bellman backup* that trains language successor function to generate possible future rollouts using only one-step samples obtained off-policy. Then, an LLM policy can be improved by refining its own suboptimal actions. This procedure improves one of the main challenges of RL fine-tuning for complex tasks—reliance on random exploration to uncover better actions–and significantly improves sample efficiency. Empirically, we show that NLAC greatly outperforms other prompting and fine-tuning

baselines on long-horizon tasks involving dialogue and tool-use. As future work, we aim to see if our approach can be combined with traditional RL policy optimization by extracting a generative scalar value from our critiques to enable more sophisticated policy improvement strategies.

## 9 REPRODUCIBILITY STATEMENT

In our work, we evaluate on existing public benchmarks for mathematical reasoning, whose datasets can be found online. We also describe in detail the implementation of our method in both Section 6 and Appendix B, including exact prompts used and hyperparameter configurations during training, so the reader can reimplement our method from scratch. Furthermore, for the camera-ready submission, we plan to open-source the code we used to conduct our empirical evaluations.

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

# A  PROOFS

## A.1  PROOF OF THEOREM 5.1

The crux of our proof is establishing that the learned representations $\Phi^\pi(s, a)$ converges to the true successor features $\Psi^\pi(s, a)$, and then using properties of successor features to link to the scalar Q-function $Q^\pi$. Let $\mathcal{R}$ be the metric space of $\mathbb{R}^d$ with the $\ell_\infty$ norm. As shorthand, we also define the iterative operator $\mathcal{T}_\Phi^\pi \Phi^\pi(s, a) = \phi(s) + \gamma \mathbb{E}_{s', a' \sim P^\pi} \Phi^\pi(s', a')$.

First, we verify that $\mathcal{T}_\Phi^\pi$ is a $\gamma$-contraction mapping on $\mathcal{R}$. For any two functions $\Phi_1, \Phi_2$, we have:

$$
\begin{aligned}
\|\mathcal{T}_\Phi^\pi \Phi_1 - \mathcal{T}_\Phi^\pi \Phi_2\|_\infty &= \sup_{s,a} \|\mathcal{T}_\Phi^\pi \Phi_1(s,a) - \mathcal{T}_\Phi^\pi \Phi_2(s,a)\|_\infty \\
&= \sup_{s,a} \|(\phi(s) + \gamma\mathbb{E}[\Phi_1(s', a')]) - (\phi(s) + \gamma\mathbb{E}[\Phi_2(s', a')])\|_\infty \\
&= \gamma \sup_{s,a} \|\mathbb{E}_{P^\pi}[\Phi_1(s', a') - \Phi_2(s', a')]\|_\infty \\
&\leq \gamma \sup_{s,a} \mathbb{E}_{P^\pi}[\|\Phi_1(s', a') - \Phi_2(s', a')\|_\infty] \\
&\leq \gamma \|\Phi_1 - \Phi_2\|_\infty\,.
\end{aligned}
$$

Since $\mathcal{T}_\Phi^\pi$ is contraction, the Banach Fixed-Point Theorem with Assumption 5.2 guarantees that minimization of our training objective in Equation 2 results in the unique fixed point

$$
\Phi^\pi(s, a) = \phi(s) + \gamma\mathbb{E}_{s', a' \sim P^\pi} \Phi^\pi(s', a')\,,
$$

which are exatly the successor features by definition.

Next, we want to establish a monotonic mapping from representations $\Phi^\pi(s, a)$ to true Q-values $Q^\pi(s, a)$. The true Q-function is defined as $Q^\pi(s_t, a_t) = \mathbb{E}_\pi[\sum_{\tau=t}^\infty \gamma^{\tau-t} r(s_\tau, a_\tau)]$. Using Assumption 5.1, and substituting the fixed-point result from earlier, we have:

$$
Q^\pi(s, a) = \mathbb{E}_\pi\left[\sum_{\tau=t}^\infty \gamma^{\tau-t}(\phi(s_\tau) \cdot w)\right] = \left(\mathbb{E}_\pi\left[\sum_{\tau=t}^\infty \gamma^{\tau-t}\phi(s_\tau)\right]\right) \cdot w = \Phi^\pi(s, a) \cdot w\,.
$$

Since the natural language values at convergence are given by $Q_L^\pi(s, a) = f_Q(\Phi^\pi(s, a))$, the monotonic mapping $g$ is defined by $g(Q_L^\pi(s, a)) = f_Q^{-1}(Q_L^\pi(s, a)) \cdot w$. $\qquad\square$

## A.2  PROOF OF THEOREM 5.2

First, we show monotonic policy improvement in each iteration of our algorithm. Recall that the refinement policy $\pi^r$ is defined as the oracle that generates $a^r$ such that $Q_L^\pi(s, a^r) \geq Q_L^\pi(s, a)$. By Theorem 5.1, a monotonic mapping $g$ exists between $Q_L^\pi$ and $Q^\pi$, and ensures that the ordering established in the language space holds in the true scalar space:

$$
Q_L^\pi(s, a^r) \geq Q_L^\pi(s, a) \implies Q^\pi(s, a^r) \geq Q^\pi(s, a)\,.
$$

Note that the policy improvement objective in Equation 4 updates $\pi$ to $\pi'$, which is an approximation of the refinement policy $\pi^r$. By definition of refinement policy, this guarantees that the new policy $\pi'$ is monotonically better than $\pi$ in terms of expected return: $Q^{\pi'}(s, a) \geq Q^\pi(s, a)$ for all $s, a$.

Next, we want to show convergence to optimal policy $\pi^*$. This naturally follows from the fact that in a finite MDP, policy iteration that guarantees monotonic improvement at every step must converge to the unique optimal policy $\pi^*$ in a finite number of iterations. Hence, at convergence, the policy satisfies $Q^{\pi^*}(s, a) \geq Q^\pi(s, a)$ for all other policy $\pi$. $\qquad\square$

# B IMPLEMENTATION DETAILS

In this section, we provide details of implementation of NLAC across the various benchmarks we evaluate. Details include the prompts used to mimic the different components of our algorithm, as well as hyperparameters configured during RL training.

Recall from Section 6 that our algorithm consists of the following novel components:

(1) **language successor model**: probabilistically generates a text prediction of what will happen to policy $\pi$ after taking an action.

(2) **language Bellman backup**: uses one-step sample of the immediate next state to also probabilistically generate a target text prediction of the future after taking an action.

(3) **language evaluator**: processes textual futures to generate a critique of an action, commenting on optimality and an explanation why by referencing potential future outcomes.

(4) **refinement policy**: uses the critique of an action to propose an improved action.

In practice, since number of futures is $k = 1$ in our experiments, we combine the successor model and evaluator into one generation by the **language critic**.

## B.1 LANGUAGE CRITIC IMPLEMENTATION

The language critic is implemented by prompting the base LLM with instruction $p_{\text{eval}}(s_t, a_t)$. In the $\tau$-bench benchmark, this is done by appending the following prompt to the history of messages comprising $s_t$ and $a_t$:

> Evaluate your last action, first predicting one possible future and then comment on whether or not your action was optimal, and if not, how it can be improved. Output should be exactly in the format:
>
> Future:
> <Predict one possible scenario of what will happen next, up to whether or not you succeed at the long-term task. Be concise and keep to a few sentences at most.>
> Optimality:
> <"Yes" or "No". If "No", explain how it can be improved in one sentence using the predicted future to justify your explanation.>
>
> Do not generate anything after the evaluation.

For a single-step task such as mathematical reasoning, the appended prompt is instead:

> For your attempted solution, please perform the following evaluation and output the result exactly in the format:
>
> Correctness:
> <"Yes or No". If "No", identify where any errors occurred. Remember the solution could be incorrect simply because the answer is not formatted correctly with the answer in the format \boxed{answer}.>

### B.2 LANGUAGE BELLMAN BACKUP IMPLEMENTATION

The language Bellman backup is also implemented by prompting the base LLM with instruction $p_{\text{eval}}(s_t, a_t, s_{t+1})$. This is done by first appending the following prompt to the history of messages comprising $s_t$ and $a_t$ to get a bootstrapped future prediction:

> The response to your latest action is (could be a tool API output or text utterance from the customer):
> {next observation}
> From this state, describe one possible scenario of what will happen next, up to whether or not you succeed at the long-term task. Be concise and keep to a few sentences at most.

Then, the target evaluation is obtained by appending the following prompt afterwards

> Evaluate your latest action. Remember your output should be in exactly the following format:
> Future:
> <Combine the observed response to your latest action with the predicted future from there, up to whether or not you succeed at the long-term task.>
> Optimality:
> <"Yes" or "No". If "No", explain how it can be improved in one sentence using the predicted future to justify your explanation.>
>
> Notes:
> 1. Do not call tools in the evaluation. They will be **ignored**.
> 2. If the action is optimal, just say "Yes" after the "Optimality:" tag and do not explain why.
> 3. Do not generate anything after the evaluation.

Another important detail when training reasoning models (that output chain-of-thought thinking by default before every generation) is that its chain-of-thought output will reference the next state $s_t$. This makes it an unsuitable training target because it references information not provided to the critic. Hence, we add an additional postprocessing step to generate a *corrected* chain-of-though thinking that removes references to such ground-truth information:

> In the above evaluation, the chain-of-thought thinking between <think>and <\think>tags likely referenced the response to your action and future, or the final score if provided.
>
> Fix the chain-of-thought thinking so that it does not refer to those quantities as a reference, but rather infers them. So instead of saying an event will happen in the future, or that the final score is 0, say that you believe it will happen.
>
> Your corrected chain-of-thought should be similar to the original in style and prose, but simply remove references to future or the final score as ground-truth information, and instead reason about how you might be able to infer future events from only the observations thus far, up to your latest action. Your output should be in to format: <corrected_think>Revised chain-of-thought thinking goes here...<\corrected_think>
>
> It is important that you enclose the corrected chain-of-thought thinking between <corrected_think>and <\corrected_think>tags, as your response will get automatically parsed by a computer. The part after the chain-of-thought thinking should be the evaluation exactly in the format described earlier.
>
> There should be exactly one <corrected_think>...<\corrected_think>block in your response. Do not include any <think>or <\think>tags within this block. Do not generate anything after the <\corrected_think>tag.

Then, we extract the corrected chain-of-thought thinking from the output and co-opt the original chain-of-thought-thinking in the target evaluation.

## B.3 REFINEMENT POLICY IMPLEMENTATION

The refinement policy is implemented by appending an additional prompt after $Q_L^\pi(s_t, a_t)$ that is the output of the language critic:

> Use the evaluation of the latest action to assess whether the latest action was optimal, and generate a revised action that fixes any problems with the latest action (can simply copy latest action if it is optimal). Output should be exactly in the format:
>
> Thought:
> <A single line of reasoning to process the context and inform the decision making. Do not include extra lines.>
> Action:
> {"name": <Name of action>, "arguments": <Arguments to the action in json format>}
>
> Note that you are outputting an action that will replace the latest one. Do not output an action that is meant to come afterwards.
>
> Do not reference the previous action or its evaluation.

Again, for LLM policies that are reasoning models, we must correct the chain-of-thought thinking that will likely reference the critique (which is not seen by the base policy). We append the following postprocessing prompt afterwards:

> In the above revised action, the chain-of-thought thinking likely used the previous action and its evaluation to guide your thinking.
>
> I want you to fix the chain-of-thought thinking so that it does not use the previous action and its evaluation as reference, but rather infers those quantites. So instead of referring to an action and its evaluation, say that if this action was chosen, then you believe the following evaluation would happen.
>
> Your revised chain-of-thought should be similar to the original in style and prose, but motivate the revised action directly from just the last observed tool or customer response, as if the revised action were your first attempt. Your output should be in to format: <corrected_think>Revised chain-of-thought thinking goes here...<\corrected_think>
> It is important that you enclose the corrected chain-of-thought thinking between <corrected_think>and <\corrected_think>tags, as your response will get automatically parsed by a computer. The part after the chain-of-thought thinking should be the evaluation exactly in the format described earlier.
> There should be exactly one <corrected_think>...<\corrected_think>block in your response. Do not include any <think>or <\think>tags within this block. Do not generate anything after the <\corrected_think>tag.

Like before, we parse the corrected chain-of-thought thinking and replace the original thinking in the output of the refinement policy.

### B.4 TRAINING DETAILS

Our fine-tuning baselines were implemented using the Volcano Engine Reinforcement Learning (verl) library (Sheng et al., 2024). We train on $8$ H20 GPU nodes, resulting in $64$ GPUs total, for a total of $30,720$ gradient steps. Training took $< 48$ hours for each benchmark. We used the following hyperparameter configuration for each benchmark, after some minimal amount of tuning:

| Hyperparameter | Setting |
|---|---|
| Maximum prompt length | 8192 |
| Maximum response length | 24576 |
| Batch size | 1024 |
| Number of iterations | 30 |
| Target network update $\tau$ | 0.005 |
| Prioritized replay buffer $\alpha$ | 0.1 |
| Optimizer | AdamW |
| Learning rate | 5e-6 |

### B.5 TRAINING THE REFINEMENT POLICY

Currently, our method relies on base reasoning capabilities for the refinement policy to generate $a_t^r \sim \pi_\theta^r(\cdot|s_t, a_t, Q_L^\theta(s_t, a_t))$ such that $Q_L^\theta(s_t, a_t^r) \geq Q_L^\theta(s_t, a_t)$. In situations where pretrained LLMs cannot refine actions through in-context learning, we describe how to explicitly train the refinement policy.

To train the refinement policy, we adopt a similar approach as Kumar et al. (2024) did to train policies to self-correct, but generalized to multi-step MDPs. Namely, for some on-policy sample $s_t, a_t, a_t^r$ from the refinement policy, we additionally train on the loss:

$$\mathcal{L}_r(s_t, a_t, a_t^r) = -\log \pi_\theta(a_t^r \mid s_t) \left( A^{\pi_\theta^r}(s_t, a_t^r) + \alpha \left( A^{\pi_\theta^r}(s_t, a_t^r) - A^{\pi_\theta}(s_t, a_t) \right) \right) . \quad (7)$$

Here, $A$ is the estimated advantage function, which is either learned (as in PPO), or obtained from averaging (as in GRPO), using Monte-Carlo rewards. We also include a bonus that is the improvement in advantage of refined $a_t^r$ over base $a_t$, where $\alpha > 0$ is a tunable parameter. Our findings indicated that this approach resulted in only marginal performance gains compared to the simpler, implicit distillation method already employed by NLAC. Furthermore, the explicit RL objective substantially increased training cost, requiring on-policy samples. Therefore, in the experiments we consider, we do not employ such training. However, in more complex tasks, this may be necessary.

### B.6 MITIGATING CATASTROPHIC FORGETTING

Several components of our method, such as the language Bellman backup and refinement policy, are never explicitly trained but merely prompted to behave according to instruction. However, a key challenge that arises is as our LLM model is trained, it becomes increasingly less competent at following such instructions. We eventually observe *catastrophic forgetting* across all our training runs, resulting in the score during training collapsing to $0$.

In our experiments, we choose to stop training before catastrophic forgetting became a noticeable issue. However, we also considered methods to minimize forgetting, notably by incorporating learning without forgetting into our training objective (Li & Hoiem, 2016). The way we do so is by introducing auxiliary loss functions that penalizes divergence from the initial, pre-trained LLM. We introduce two additional losses for the language Bellman backup and refinement policy, respectively:

$$\mathcal{L}_{\text{lwf},1}(s_t, a_t, r_t, s_{t+1}) = D_{\text{KL}} \left( \mathcal{B}_L M_\theta(\cdot \mid s_t, a_t) \,||\, \mathcal{B}_L M_{\text{init}}(\cdot \mid s_t, a_t) \right) \quad (8)$$

$$\mathcal{L}_{\text{lwf},2}(s_t, a_t, Q_L(s_t, a_t)) = D_{\text{KL}} \left( \pi_\theta^r(\cdot \mid s_t, a_t, Q_L(s_t, a_t)) \,||\, \pi_{\text{init}}^r(\cdot \mid s_t, a_t, Q_L(s_t, a_t)) \right)$$

Ultimately, we found that such objectives were able to prevent collapse, allowing our LLM model to be trained for more iterations. However, training for more iterations did not prove significantly advantageous in improving final performance. Therefore, we did not include such objectives in our default algorithm.

# C QUALITATIVE EXAMPLES

---

…
**Agent:**
Thought: We have identified the object is an edible food item. I should split the food category into fruits, vegetables, grains, meats, etc.
Action: *is the object a fruit?*
**User:** YES
…
**Agent:**
Thought: The object is a non-red fruit. To narrow down the type of fruit, I can ask about its preparation to separate common meal ingredients from desserts or snacks.
Action: *is the object commonly found in a salad?*
**User:** YES

**Agent (base):**
Thought: The object is a non-red fruit often found in salads. I will ask about the next most common color.
Action: *is the object typically green in color?*

**Critique:**
Future: The agent will eventually guess the color by searching through fruit colors. However, it is possible that color is not the optimal characteristic and size of taste are required to identify the fruit.
Optimality: No. Linearly searching through fruit colors misses key discriminators like taste and size.

**Agent (refined):**
Thought: I should ask about whether the fruit is used in sweet or savory contexts.
Action: *is the object commonly found in a fruit salad?*

---

Figure 3: Sample timestep on 20Q, where the LLM agent attempts to guess the hidden object "raisin." The base LLM agent has narrowed down the object to a non-red fruit found in salads, but proceeds to search over the color. However, color is often not the most defining characteristic, so it is more optimal to search over other discriminators such as taste or size.

---

**User:** My name is Sofia Thomas. I'd like to inquire about making a **couple of exchanges**.
**Agent:**
Thought: *To help Sofia modify her order, I first need to authenticate her. Since she provided name and zip code, I should use find_user_id_by_name_zip to get her user ID.*
Action: find_user_id_by_name_zip("Sofia", "Thomas", "75307")
…
**Agent:**
Thought: *The product details show there's an available robotic vacuum with pet hair removal … I need to confirm this new item with the user and ask for a payment method to cover the difference.*
Action: respond("We have an available vacuum cleaner…Please confirm if you want this new item")
**User:** Yes that works for me. The payment method ID is paypal_5334408.

**Agent (base):**
Thought: *I need to call the modify_pending_order_items tool with the order id, item ids, new item ids, and payment method id to complete the modification…*
Action: modify_pending_order_items("#W2297866", ["7407609582"], ["4965355367"], "paypal_5334408")

**Critique:**
Future: The modify_pending_order_items tool will successfully update the first order. However, when the user requests to modify another item, the agent will attempt to call modify_pending_order_items again, but this will fail as it is no longer modifiable.
Optimality: No. The action does not anticipate the need for subsequent modifications.

**Agent (refined):**
Thought: *I should confirm with the user that no other items need modifications.*
Action: respond("Before I make the modification, please confirm that there are no other orders you wish to modify.")

---

Figure 4: Sample timestep on $\tau$-bench where a base LLM agent fails by modifying the database (which can only be done once according to the guidelines) when more exchanges are likely needed. The natural language critic correctly identifies why the action is suboptimal, and explains it in language so that the same LLM can process the critique and correct its action.

