# OpenReview forum: "Natural Language Actor-Critic: Policy Iteration in Natural Language Space"
_ICLR.cc/2026/Conference — Submitted to ICLR 2026_

### Official Review · Reviewer_JPr6 · 2025-10-20

**Soundness:** 4
**Presentation:** 3
**Contribution:** 3
**Rating:** 4
**Confidence:** 3

**Summary:**

The paper presents a way to train LLM agents using an actor-critic approach where the critic provides a natural language critique of the agent's action together with an action refinement proposal. The critique is generated by training a language successor model to predict the outcome of the forthcoming sequence of actions. The authors took care to have the whole process implemented as off-policy RL to improve sample efficiency, and used a prioritized replay buffer ranked on the prediction loss to improve learning efficiency. The approach is evaluated using two LLMs of the Qwen family against React, fine-tuning and RL baselines, and shown to perform well.

**Strengths:**

- the point that scalar-based, standard RL relies on blind exploration and that LLM knowledge can be used to refine actions more efficiently is a good point
- the proposed architecture is sound, and most of its formalization is clear. See below for a critique on this point
- the results are good (although they rise a few questions, see below)
- the paper is reasonably well-written and easy to follow

**Weaknesses:**

- I would like to see a comparison of performance against closer baselines such as NLRL (Feng et al. 2025, NeurIPS), PNLC (Hong et al. 2025) or ArCHer (Zhou et al. 2024b). In particular, the NLRL paper has a NLAC algorithm (see Algo. 3 p. 19 and Section 4.3) which is very close to this one, thus a detailed comparison of the methods and an experimental comparison are mandatory. This is the main weakness of this work.
- it is not specified whether the results are obtained just from one run, so their significance should be questioned.
- clarity would benefit from a figure showing the architecture with all the components: the policy, the language successor model, the language evaluator, the refinement policy, the replay buffer, ... and how they are connected.
- though I'm not expert of this literature, it seems to me that the main difference to NLRL, which aslo has an NL actor-critic architecture, is in the refinement policy component. And to me this refinement policy component is the one whose formalization needs the most improvement

**Questions:**

- an argument in favor of NLAC is computational and sample efficiency. Could you provide training and inference times for your experiments, and compare them to the ones that would be required to use NLRL, PNLC or ArCHer?
- could NLRL, PNLC or ArCHer be applied to your benchmarks and if not, why? If yes, can you give the results for at least one of them? If not, could you provide the performance of your work in the benchmarks used in the NLRL paper?

- line 193: "We believe that the key for a critic to derive these explanations is the prediction and analysis of future outcomes." -> Did you try alternative mechanisms and compared their performance to yours? Did some other papers do so? You should turn this "belief" into a scientifically established fact.

- Eq. (1), the notation is unclear. Is Pr[] a probability. Rather than Pr[x], y, z, shouldn't you use P(x|y,z)?

- About Assumption 4.1: why is it formally necessary to make the process sound? Besides, what happens if two critics correspond to the same scalar?

- I'm not convinced by the formalization of the refinement process calling upon a target policy and projection to the base policy. What I understand is that the LLM is just asked once through a prompt to propose a refinement based on the critique.

*Remarks*:
- having "policy iteration" in the title is weird, as your work is truly an actor-critic (AC) approach. AC is the equivalent of PI when the transition function is unknown...
- paragraph "Reinforcement Learning" in Section 3 suffers from two slight over-simplifications: First, policy gradient methods such as PPO do not just use the gradient $\nabla_\pi$, they use it mutiplied by a value, action-value or advantage function which is a kind of critic. So you need a critic in these methods too, even though they do not qualify as actor-critic. Second, your action space being discrete, rather than using $Q(s_{t+1}, \pi(s_{t+1}))$ as in DDPG, TD3 and SAC, you may use $max_a Q(s_{t+1}, a))$. So the paragraph could be sligtly reformulated to highlight that there are other options.
- it would be nice if hyper-params k and m could appear in Algo 1
- in the experiments, you could say once and for all that success is rewarded with +1 and speak about success in task descriptions. BTW, you don't say anything about failure, I assume the reward is 0 but you should specify it.
- line 408 I had to search what k was (I remembered for m). Please remind the reader...
- line 454: "The only task where NLAC matches other methods": you mean the only task where the performance of NLAC is not superior to other methods (we may understand that it is worse on all others :)).
- results on 20Q are given as percents (between 0 and 100) whereas in $\tau$-bench they are given between 0 and 1 -> you could use the same normalization
- acheiving a 30% improvement: given (win or success) rates, I would rather say that NLAC is +~9% over competitors on 20Q and +~10% over competitors on Retail (rather than take the percentage over the competitors performance).


typos and minor remarks:
- line 148: use \citep for Yao et al. 2022.
- line 176: "based off of evaluations". You don't mean based on evaluations, right? So rather say what it is based on.
- line 185: "LLMs are better suited to process and generate natural language over scalars" -> you mean "than scalars"?
- line 186: "Therefore, we believe evaluation that is in natural language space leverages prior text-based reasoning capabilities
of LLMs, and thus will largely improve sample efficiency." -> you could refer to Fig. 1 again here.
- line 191: we will discuss later -> we discuss later
- line 192: ultimately incredibly -> avoid too strong adverbs. If this is incredible, should I believe you? :)
- line 213: some semblance -> ressemblance?
- line 223: will "discount" -> "discounts"
- "Equation 3" -> Equation (3) (use \ eqref { }). Do this everywhere.
- line 290: using $\mathcal{M}$ and M to denote two different things is not a good idea
- line 333: interpretted -> interpreted
- line 333: "but using generations by the refinement policy rather (than?) a teacher policy that is a separate model." -> I don't understand what you mean
- line 400: as well an -> and an (?) or as well as an (?)
- line 406: on any airline scenario(s).
- line 480: signficantly -> missing i
- the Carta et al. paper is given two identical references

---

> ### Author Response · Authors · 2025-11-17
> **Author Response to Reviewer JPr6 (Part 1 of 2)**
>
> Thank you for your review, as we found it incredibly helpful for improving the paper. We have updated the paper based on your feedback and provide a summary of updates below. Then, we will address in more detail each of your concerns.
>
> **Updates to the paper**
>
> We have made several updates to the paper to improve clarity as well as bolster our results:
>
> (1) We improved our discussion about how our method differs from NLRL and added comparisons in the experiments. We will discuss this in more detailed below, but on a high-level, NLRL requires enumerating in-context over environment dynamics (in the form of single or multi-step on-policy rollouts) and over possible actions, then getting aggregated predictions, which hinders scalability to general tasks; our method NLAC aims to circumvent these issues by additionally learning a language successor function that probabilistically generates textual futures, and using these futures to iteratively refine actions.
>
> (2) We added a section on theoretical analysis showing our method can converge to the optimal policy. The key insight we leverage is that when learning our successor function, our representations converge to the unique **successor features** of the underlying MDP [1]. This gives us a way to connect our learned language values to the true scalar Q-values.
>
> (3) We added figures showing how all the different components of our method connect during both policy evaluation and policy improvement.
>
> **Comparison to NLRL**
>
> You are correct that NLRL is the closest related work, and our method NLAC can be considered an instance of the paradigm introduced by the NLRL work. As we will discuss, there were key details about NLRL that we initially believed made it intractable for our experiments, but we have hence changed our opinion and compared against it.
>
> To our knowledge, NLRL is the first work to introduce learning a language critic for RL optimization. However, their proposed algorithm had two drawbacks that made it difficult to scale to general LLM agent tasks, which our method NLAC aims to address:
>
> (1) During policy evaluation, the critic in NLRL is trained by aggregating multiple on-policy samples in-context and generating a holistic evaluation using these samples. Such on-policy samples can simply consist of next state with future predicted value (like in standard temporal difference learning), but for complex tasks, we found enumerating over environment transitions in-context for one training sample to be too expensive. In NLAC, rather than training the critic to output aggregated evaluations, we train a successor model to probabilistically generate rollouts. We found that such training can be done using a single transition like in traditional RL, is theoretically motivated by learning successor features, and can later be used for action refinement.
>
> (2) During policy improvement, NLRL provides multiple candidate actions and their evaluations and performs an "argmax" over them in natural language space. In our settings, the action space is too large for such enumeration to be tractable, and we found empirically that simply sampling a tractable subset of actions does not usually adequately explore the action space. Hence, in NLAC, we consider iterative refinement of the policy. In order to refine the policy, the evaluation must explain how an action can be improved, which we provide by including a description of what will likely happen afterwards.
>
> In the updated paper, we compare against NLRL, where we limit to 8 in-context transitions and actions to keep training tractable; note that this can be interpreted as providing 8 times more samples during training. NLRL performs competitively in both math reasoning and 20Q, but falls short in \tau-bench. We found that in \tau-bench, which has the most complex dynamics and action space, the language values learned by NLRL were always positive in sentiment. We believe this could be due to not seeing enough in-context samples during training to overcome the implicit positive bias of instruction-fine-tuned LLMs.

---

> ### Author Response · Authors · 2025-11-17
> **Author Response to Reviewer JPr6 (Part 2 of 2)**
>
> **Significance of results**
>
> Our empirical results were from averaging over 3 independent training runs. It was our mistake that we omitted this detail and have corrected it in the revised paper.
>
> **Sample efficiency of NLAC**
>
> We did find that NLAC was able to converge to higher validation score with fewer samples than PPO or GRPO. We will update the paper with training curves to show this fact. We find that measuring training time is difficult as it is too dependent on the implementation details of the methods, and the response times of API calls to the LLM acting as the environment.
>
> **Not convinced by refinement**
>
> You are correct that our policy improvement relies on the LLM being able to propose a better action using the textual evaluation. We found this to be true in the tasks that we consider due to our empirical results. However, in an earlier version of our algorithm, we did train the refinement policy using PPO on the improved score (similar to how it was done in the ScoRe paper [2]). In theory, this guarantees that the refinement policy will find a better action. However, we ultimately excluded this step because it compromised the ability of our algorithm to be trained off-policy, while not providing significant improvements in performance. In the following days, we will include in the Appendix a discussion of this additional training.
>
> **Usefulness of successor function**
>
> It is true that other alternative mechanisms exist to provide the textual evaluation with signal on how to improve the action. The reason why we ultimately chose the successor function was because the resulting objective was very principled. Specifically, our language Bellman backup looks like the language analogue of how environment models were trained in model-based RL [3]. This enables us to perform theoretical analysis of the effectiveness of our algorithm, which we have included in our revised paper.
>
> [1] https://arxiv.org/abs/1606.05312
>
> [2] https://arxiv.org/abs/2409.12917
>
> [3] https://arxiv.org/abs/2010.14496

---

> > ### Comment · Reviewer_JPr6 · 2025-11-17
> > **Quick reactions**
> >
> > I really appreciate the author's effort to answer my points quickly to give me an opportunity to provide more feedback. I understand that they would need more time to provide more satisfactory answers, but they are already doing a great job.
> >
> > I won't have time to fully read the paper again before leaving. A few points:
> >
> > - for the next revised version, I suggest the authors to highlight in a different color the changes they made with respect to the previous version, so that reviewers don't need to read everything again
> > - the list of typos I provided have not been taken into account in the current revised version
> > - the two figures added to the revised version need improvement to be more readable and increase their informative value
> > - more importantly, from the authors response to my review and to the NLRL authors' points, my feeling is that the paper is improving as most comments are going in the right direction. So if after the full rebuttal process the other reviewers decide that the paper deserves acceptance, I'll be happy with their decision.
> >
> > - about policy refinement, a related work that I forgot to mention in my initial review is the following: Yang, R., Ye, F., Li, J., Yuan, S., Zhang, Y., Tu, Z., Li, X., and Yang, D. (2025) The lighthouse of language: Enhancing LLM
> > agents via critique-guided improvement. arXiv preprint arXiv:2503.16024
> >
> > Once again, I apologize for not being able to discuss the submission further, and I wish everybody a fruitful rebuttal process.

---

### Official Review · Reviewer_MUW2 · 2025-10-29

**Soundness:** 3
**Presentation:** 3
**Contribution:** 3
**Rating:** 6
**Confidence:** 4

**Summary:**

The paper "Natural Language Actor-Critic: Policy Iteration in Natural Language Space" proposes a principled actor-critic method, that uses a finetuned LLM either as actor and critic. The novelty lies mainly in the use of a generative critic, that produces feedback predictions on the future of trajectories, rather than -- less-informative -- numerical rewards. The approach takes inspiration from fundamental works in RL, specifically works about the successor measure, and nicely transfer them in the agentic setting with natural language feedbacks. Experiments demonstrate the good behavior of the approach.

**Strengths:**

- Interesting paper that takes inspiration from strong fundations. While optimizing with LLMs is always a bit opaque and lacks guarantees, I feel the paper makes one useful step toward more principled LLM agents finetuning.

- Good experimental results

**Weaknesses:**

- Presentation :  While writting is globally rather good, I was sometimes a bit lost during section 5 reading. The structure and discussion could be improved from my point of view. A starting point would be to specify more clearly that $\cal M$ is a central finetune LLM that is used from various prompts to cover different goals building the various building blocks of the method. Better formally defining its inputs and outputs would beuseful.

- Training objective : Finally, the model $\mathcal{M}$ is used for all components of the method (with different prompts as input), but it is \textit{fine-tuned} only for two tasks: (1) matching the successor measure, using $p_{\text{pred}}$ as input, and (2) maximizing the likelihood of the refined action, using $p_{\text{react}}$ as input. From what i understand, there is no additional training for other types of tasks --- these rely solely on the general capabilities of the LLM, elicited through well-crafted prompts. No regularization term is applied either, which would avoid deviating too far from the reference model, possibly inducing catastrophic forgetting of the model’s general abilities. From my point of view, this issue could occur if the model is fine-tuned too aggressively, making the training process difficult to stabilize. More details on this aspect would be valuable.

- Performance analysis : The paper only reports final accuracies for each considered task, but does not analyze the training dynamics. Performance curves regarding various metrics during the training process would be useful to understand these dynamics. Is the model oscillating ? Is it deviating far from the reference model ? How the entropy of the model evolves regarding the different inputs ? How does training on for instance the successor measure prediction transfer for other tasks ? etc.

**Questions:**

Beyond questions / concerns reported in the weaknesses section, I have the following additional questions :

- Authors consider SAC as a baseline. But not enough is said about how it used. Are the actor and the critic still LLMs that you finetune ? If yes how. If no, how does it deal with textuel inputs and actions ? Also, it is known that SAC is not good for the discrete domain. Wouldn't it be preferable to consider more common RL methods for LLMs RL finetuning, such as PPO / GRPO / DPO ?

- In many places in section 4, author say that the only network to be finetuned is the successor model. But in section 5 I understand that finally not, as the same model is used for many tasks (and is also finetuned for the policy task). Please clarify

- $\theta$ appears at different place in Equation (6). The gradient is only backpropagating through the policy (the first term) right ? Not through $Q^\theta$ for instance ?

- Prioritized replay is said using a alpha parameter. But nothing is said about is. It is a temperature temperatur regarding errors on transitions ? or a factor of Importance Sampling as done in the original paper ? what kinds of errors are used for this priorization (the successor measure divergence or the negloglikelihood of the selected action ?

- An interesting abblation would be to directly use the refined action as the sampled action, rather than requiring a policy component, with its associated loss. Is it actually useful ?

- Eq (2) uses a D_f term without discussing it at any place. We understand that this a kind of f-divergence but it could be useful to precise it .
- Defining 4.3 that tells that E takes sequences of rollouts associated with rewards as input is not fully consistent with its formalization in eq (3)



- In section 5, formalization of the language bellman backup : P_L ==> B_L

---

> ### Author Response · Authors · 2025-11-27
> **Response to Reviewer MUW2**
>
> Thank you for your review. You offered some suggestions that we have taken into account in our revised paper (see joint response above for all updates). We also aim to address your individual points below.
>
> **Potential for catastrophic forgetting**
>
> You are correct that some aspects of our method rely on the base reasoning capabilities of the LLM via prompting: (1) deriving the target language Bellman backup as a combination of immediate next state and bootstrapped future description, and (2) proposing a refined action using a previous action attempt and its evaluation.
>
> Back when developing our method, we did consider catastrophic forgetting as a potential issue with our approach and experimented with alternatives. In our revised paper, we add two new subsections explaining modifications that we considered making to our algorithm, namely: (1) explicitly training the refinement policy using policy optimization on achieved reward after refinement, and (2) regularization to the initial LLM model to mitigate catastrophic forgetting. Overall, we excluded these modifications from the default implementation of our method because we found that though they improved stability (allowing for training for more iterations), they ultimately did not improve final performance by a significant enough margin. Specifically, though catastrophic forgetting eventually occurs, it was not a noticeable issue on the relatively low sample regime that we train on. However, we still explain these modifications in detail as they could be more necessary in different tasks or base models than the ones we evaluate one, as such changes could affect when catastrophic forgetting becomes an issue.
>
> **Performance analysis**
>
> This is a completely valid point that final accuracy may not give enough signal to evaluate a method. To address this, we also add learning curves to our revised paper showing the validation accuracy of our method across 3 independent runs, and compare it to traditional RL fine-tuning using PPO. The curves show that our method converges in fewer iterations and is more stable than PPO.
>
> **Answers to questions**
>
> Finally, you raised some important questions that we will answer below:
>
> 1. The SAC baseline that we evaluate against is one possible ablation of our approach, where the critic outputs a scalar instead of a textual evaluation. Implementation-wise, it is similar to PPO, in that a separate policy and critic LLM are learned, but the policy is learned via projection to the soft Q-values $\pi(a | s) \propto \exp(Q(s, a))$ like in SAC. We agree that SAC may not be optimal for our experiments, but we also evaluate PPO/GRPO, which are more popular algorithms for the tasks we evaluate on. We evaluate SAC in addition to PPO/GRPO simply because its policy projection step is more similar to what our method NLAC does, making SAC a middle-ground between the two paradigms of algorithms.
> 2. Our method requires two training objectives on the same base LLM: for (1) policy evaluation via fitting to the language Bellman backup, and (2) policy improvement using the refinement policy. When we say only the successor model is fine-tuned, we were referring only to the policy evaluation component of our method (i.e. the actual critic that generates evaluations using the successor model is not trained). However, the policy is still fine-tuned for the policy improvement component. We made this distinction more clear in our revised paper.
> 3. You are correct in that the gradient only propagates through the policy term, so our objective can be interpreted as distillation from the refinement policy. More recent experiments of ours consider training on the reverse direction of KL-divergence instead, but they are currently reserved for future work.
> 4. For the prioritized experience replay, we use $\alpha$ to scale the errors of transitions during sampling, and $\beta$ to scale the importance sampling weight. We follow the definitions for $\alpha, \beta$ from the original paper. We clarify this detail in our revised paper in case other readers are unfamiliar with how these variables are used.
> 5. We considered directly using the refined action, and found that it performed marginally better than using the refined action to distill our base policy. However, such refinement would incur a much higher inference cost (at least 2x more), so we ultimately decided the increased cost did not justify the slight gains in performance.

---

### Official Review · Reviewer_6UAb · 2025-11-01

**Soundness:** 2
**Presentation:** 2
**Contribution:** 2
**Rating:** 4
**Confidence:** 4

**Summary:**

This paper proposes Natural Language Actor-Critic (NLAC), a new actor-critic algorithm designed to train Large Language Model (LLM) agents. The authors identify that standard policy gradient methods are often unstable and sample-inefficient for long-horizon tasks with sparse rewards. To address this, NLAC replaces the traditional scalar-value critic with a generative LLM critic that produces textual critiques of actions. This natural language feedback aims to provide a richer, more actionable training signal by explaining *why* an action is suboptimal. The critic is trained off-policy using a novel "language Bellman backup", and the actor (policy) is improved by using the critic's textual evaluation to refine its own actions, which are then used as targets for a distillation-style update. The authors demonstrate that NLAC outperforms existing methods like PPO, GRPO, and RFT on tasks involving reasoning, dialogue, and tool-use.

**Strengths:**

- Important Problem: The paper tackles a critical and timely problem. Improving the stability and sample efficiency of RL fine-tuning for LLM agents, especially in complex, long-horizon scenarios with sparse rewards, is a significant challenge in the field.

- Well-Motivated Method: The central idea of using language-based evaluations is strongly motivated. Instead of a simple scalar value, a natural language critique can provide richer, more interpretable feedback, which intuitively leverages the inherent reasoning capabilities of LLMs for policy improvement.

- Data Efficiency: The algorithm is designed to be trained off-policy, which offers a promising and more data-efficient alternative to the prevailing on-policy methods (like PPO) that require new trajectories at every step.

**Weaknesses:**

- Soundness of Policy Improvement: The policy improvement step, defined in Equation 6, appears to have a potential flaw. The algorithm trains the base policy $\pi_{\theta}$ to mimic the refined action $a_t^r$. This update relies on the crucial assumption that the refined action $a_t^r$ is guaranteed to be better than the original action $a_t$. However, this improvement is not enforced; the refinement policy $\pi^r$ is merely *prompted* to generate a better action.

- Limited Empirical Evaluation: The experimental validation could be more robust.

  1. Model Diversity: The paper only reports results on two models from the same family (Qwen2.5-7B-Instruct and QwQ-32B). Testing the method on other widely-used and structurally different open models (such as Llama, or DeepSeek-distilled models) would be necessary to demonstrate the generalizability of the approach.

  2. Lack of Ablations: The paper introduces several novel components, chiefly the language Bellman backup objective ($\mathcal{L}_1$) and the refinement-distillation objective ($\mathcal{L}_2$). However, there are no ablation studies to isolate the contribution of each component. For instance, how does $\mathcal{L}_1$ compare to a simpler critic trained on Monte Carlo rollouts? How crucial is the $\mathcal{L}_2$ update? Without these ablations, it's difficult to attribute the performance gains to specific parts of the proposed algorithm.

**Questions:**

- Regarding the primary weakness: What happens if the refinement policy $\pi^r$ consistently fails and generates an action $a_t^r$ that is *worse* than $a_t$? Since the policy $\pi_{\theta}$ is trained to mimic $a_t^r$ regardless, how does the algorithm prevent or recover from this "actor-critic-refiner" misalignment?

- The paper claims better data efficiency but provides results based on a fixed number of gradient steps. Could the authors provide comparisons based on the number of environment samples or wall-clock time? How does the significant computational overhead of generating critiques and refined actions compare to the standard rollouts of PPO?

- The practical implementation relies on complex, multi-step prompting, especially for correcting the chain-of-thought in the Bellman backup and refinement policy. How sensitive is the algorithm's performance to the precise wording and structure of these prompts?

---

> ### Author Response · Authors · 2025-11-27
> **Response to Reviewer 6UAb**
>
> Thank you for your review. We took into account many of our concerns and suggestions to improve our revised paper (see joint response for more details). We will also address each individual point you made in your review below.
>
> **Soundness of policy improvement**
>
> That is a valid concern with our method, as we currently rely on the base reasoning capabilities of the LLM in order to propose refined actions. This was a consideration we had while developing our method, and we considered two possible modifications: (1) explicitly training the refinement policy using policy optimization on achieved reward after refinement (similar to ScoRe [1]), and (2) regularization to the initial LLM model to mitigate catastrophic forgetting. In our revised paper, we go into more detail about both modifications. Ultimately, we found that such additions did not improve performance by a significant amount, while incurring higher training cost. However, we agree with the reviewer that relying on the base LLM may not be feasible for all tasks and models, so we include discussion on how to perform this additional training as a possible addition to our base default algorithm.
>
> In addition, we realize that our initial submission did not perform an theoretical analysis on the soundness of our overall algorithm. Therefore, in our revised paper, we add a new section where we derive that the representations learned by our method converge to successor features of the underlying MDP, which gives us guarantees of monotonic policy improvement and convergence to the optimal policy.
>
> **Lack of ablations**
>
> This is a good point, as we do not compare our method NLAC against the simpler method of just aggregating Monte Carlo rollouts. A lot of considerations in designing our algorithm were for tractability and practicality. Namely, our language Bellman backup circumvents requiring multiple on-policy rollouts via a temporal-difference objective, and our refinement objective avoids enumerating the action space in order to perform an argmax over them; we believe this is important for our algorithm to be scalable to general LLM agent environments with complex dynamics and combinatorial action spaces.
>
> To address your concern, we add a comparison to a new ablation NLRL [2], that is a precursor to our method that precisely learns a language value function by aggregating on-policy samples, and performs an explicit maximization over actions rather than refinement. We initially did not compare because we thought NLRL would be difficult to scale to our experimental domains, but we admit such comparison is important and evaluate against a version that is limited to 8 on-policy samples rather than full enumeration (see joint response for more details). NLRL performs competitively in both math reasoning and 20Q, but falls short in \tau-bench. We found that in \tau-bench, which has the most complex dynamics and action space, the language values learned by NLRL were always positive in sentiment. We believe this could be due to not seeing enough in-context samples during training to overcome the implicit positive bias of instruction-fine-tuned LLMs.
>
>
> **Better sample efficiency**
>
> In our revised paper, we add learning curves of our method against standard online policy optimization. The learning curves show that our method reaches peak validation score in much fewer gradient steps than PPO (around 40% fewer steps). We agree with the reviewer that generating training samples will be more expensive due to being the language space rather than scalars. However, our method works off-policy, so we only  generate 1/4 of the samples required by PPO for the same number of gradient steps; all generated samples are stored in a replay buffer and can be used for multiple gradient steps.

---

### Official Review · Reviewer_mvfc · 2025-11-01

**Soundness:** 3
**Presentation:** 2
**Contribution:** 2
**Rating:** 4
**Confidence:** 3

**Summary:**

Training LLM agents for long-horizon, multi-turn tasks with sparse rewards is unstable and sample-inefficient under on-policy policy-gradient methods (e.g., PPO/GRPO).
While standard actor–critic could, in principle, be applied, the authors argue it fails to exploit pretrained LLMs’ text-based reasoning.

They propose Natural Language Actor-Critic (NLAC), which replaces a scalar critic with a generative, natural-language critic.
The critic produces textual evaluations of actions that the policy can read and use for self-refinement, enabling policy iteration in language space and training that can proceed off-policy.

Concretely, NLAC defines a language successor model $M^\pi$ and a language Bellman backup $\mathcal{B}_L$ that constructs one-step, off-policy targets over distributions of future textual descriptions (and terminal reward).
A language evaluator $E$ then aggregates one or more sampled futures from $M^\pi$ in-context to produce a textual critique $Q^\pi\_L$.
Conditioned on this critique, a refinement policy proposes an improved action $a\_t^r$.
The base policy is then distilled toward these refined actions via a likelihood/divergence objective (rather than enumerating actions or using scalar Q for a closed-form policy).

Experiments on MATH500-Hard, 20 Questions (20Q), and τ-Bench show that, under a fixed training budget of 30,720 gradient steps, NLAC consistently outperforms RL fine-tuning baselines (e.g., PPO/GRPO) and surpasses GPT-4.1 prompting on the long-horizon tasks.

**Strengths:**

- The proposed critic can explain why/how an action is suboptimal, reducing reliance on random exploration in language action spaces.
- The authors reuse a single LLM with role-specific prompts (policy, successor, evaluator, refiner), simplifying engineering.
- NLAC beats PPO/GRPO and even GPT-4.1 prompting on multi-turn dialogue/tool-use benchmarks.
- The paper extends actor–critic ideas into language space—an interesting perspective for PRMs and distributional RL–inspired methods.

**Weaknesses:**

- Lacks contraction/monotonic-improvement or SAC-style guarantees; the language backup remains heuristic.
- Limited ablations (e.g., **numbers of sampled futures $k$ and refinement rounds $m$**; **further gradient steps**; reverse-KL vs. forward-KL; the effect of prioritized replay; EMA rate; and smaller base model than 7B).
- It is unclear how to apply the approach to general RLHF tasks with non-binary and/or dense rewards.

**Questions:**

- (weakness 1) Since the proposed approach is fairly complex, do the authors provide any theoretical derivations to justify such a complex design?
- (weakness 2) Do the authors have plans for additional ablation studies?
- (weakness 3) Do the authors have plans to apply NLAC to general RLHF tasks with non-binary and/or dense rewards?
- I believe a similar high-level concept paper [1] exists (with different details). Could the authors compare their method against [1]?
- Are there any GRPO results with larger sample counts (e.g., 32 or 64)? Using only 4 seems small given GRPO’s motivation.
- What is the exact role of the scalar reward in this paper? Is it important for performance? Are there any ablation studies on this as well?

[1] Wang, Hanyang, et al. "Text2Grad: Reinforcement Learning from Natural Language Feedback." arXiv preprint arXiv:2505.22338 (2025).

---

> ### Author Response · Authors · 2025-11-21
> **Response to Reviewer mvfc**
>
> Thank you for your review. We believed that you raised valid concerns about our work that we aimed to address in our revised paper, which we will summarize below.
>
> **Lack of theoretical derivations**
>
> We completely agree that such derivations are important, simply to show that our training objectives are actually principled. We initially were not sure how to proceed, as traditional analyses of actor-critic algorithms require showing convergence to the true scalar Q-values, which we never try to learn.
>
> However, a key insight we realized is that we can show convergence of our learned representations (used to generate the textual evaluations) to the unique successor features of the MDP. This gives us a way to connect our learned language values to the true scalar Q-values. In this manner, we are able to show eventual convergence to the optimal policy, similar to SAC.
>
> **Additional ablation studies**
>
> We have added an additional comparison to a prior method NLRL, that can be considered an ablation of our method that also learns to output textual rather than scalar evaluations, but via different training objectives that we believed were not as scalable. This method is similar to the Text2Grad paper that you mentioned, but generalizes to multi-step LLM agent tasks (whereas to our knowledge, Text2Grad is proposed only for single-step tasks such as question-answering that are solved via RLHF). See our joint response for a detailed discussion of this comparison.
>
> Furthermore, we also improved our discussion of various modifications that can be made to our algorithm that includes: (1) adding regularization to the initial policy to allow for training for further gradient steps, (2) increasing value of hyperparameters $m$ and $k$, and (3) changing the sampling strategy from the replay buffer. Ultimately, our evaluations showed that these modifications decreased the efficiency of our algorithm without providing significant improvement to the final performance.
>
> **NLAC for non-binary/dense rewards**
>
> This is an important clarification that we made in our revision. Our method NLAC is not limited to binary or sparse reward settings. Namely, the textual evaluations that we learn can be applied to any reward function, as the evaluation only comments on how good an action is, and tries to explain why by analyzing future outcomes. The reason our experimental settings considered sparse, binary rewards is simply because the overwhelming majority of publicly available benchmarks adopt this style of reward function.
>
> **Answers to remaining questions**
>
> Finally, the reviewer raised some important questions that were not addressed above:
>
> 1. Our choice of number of samples for GRPO was determined by hyperparameter search. We found that increasing the number of samples per problem actually harmed performance, as it meant less diversity of unique problems in one training batch.
>
> 2. The exact role of reward in the language Bellman backup is to ground the textual evaluation. Without reward, there is no explicit signal of whether or not a future predicted by the successor function was desirable. This grounding is analogous to how reward is used in the training of scalar Q-values. Hence, we do not think it would make sense to remove the reward entirely, as then there is no ground-truth label for how good a trajectory was. Note that in our evaluations, the reward is sparse, so it is only relevant when training on the last timestep of a trajectory.

---

### Public Comment · ~Xidong_Feng1 · 2025-11-12
**Concerns Regarding Substantial Overlap and Attribution of Prior Work NLRL (arXiv:2411.14251)**

We are the authors of the paper "Natural Language Reinforcement Learning" (NLRL), first posted on arXiv in Nov 2024 (https://arxiv.org/abs/2411.14251). We thank the reviewers for their diligent work and would like to raise serious concerns regarding the substantial overlap between this submission, "NATURAL LANGUAGE ACTOR-CRITIC" (NLAC), and our prior work.

We especially appreciate Reviewer JPr6 for correctly identifying our paper as a critical and missing baseline. We believe a closer examination shows that the similarities extend beyond a missing comparison and touch upon the core conceptual framework, methodology, and even textual phrasing presented in this submission.

Our key concerns are as follows:

**Identical Core Contribution:** The central idea of reformulating RL principles into a natural language space for LLMs is the foundational premise of both our work and this submission. NLAC's "natural language critic" is functionally identical to the "Language Value Function" that we introduced and formalized in NLRL.

**Systematic Rebranding of Key Concepts:** The core theoretical components of NLAC appear to be direct counterparts to those in our work, but have been renamed. This includes the "Language Bellman Backup" (our Language Bellman Equation), "Refinement Policy" (our Language Policy Improvement), and "Language Successor Model" (part of our Language MC/TD formulation).

**Strikingly Parallel Structure and Phrasing:** The similarities extend to the structure and specific phrasing of the methodology. For instance, the introduction to the "Policy" component in NLAC's Section 5.1 almost exactly mirrors how we introduce "LLMs as language policy" in our work (NLRL v1, Sec 4.1). Both sections begin by stating that LLMs are used as policies, immediately highlight the benefit of leveraging chain-of-thought, and cite the exact same foundational papers (Wei et al., Yao et al.) to support this point. This degree of structural and textual parallelism is highly unusual and suggests more than an independent derivation of ideas.

**Mischaracterization of Our Work to Claim Novelty:** This is a critical point of concern. The NLAC paper (lines 122-125) incorrectly claims our work is limited to on-policy Monte Carlo methods. This is a factual misrepresentation. Our paper explicitly formulates and discusses both on-policy "Language Monte Carlo" and, crucially, "Language Temporal-Difference" updates (see Section 3.2.1 and Equation 6 in v3). NLAC then presents its "language Bellman backup" (an off-policy TD update) as a novel solution to a limitation that does not exist in our work.

**Omission of Direct Experimental Comparison:**: As noted by Reviewer JPr6, a direct comparison is mandatory. Given the profound conceptual similarity, the absence of NLRL as a primary baseline makes it difficult to assess the true scientific contribution of NLAC.

For verification, our paper is publicly available on arXiv:
v3 (latest): https://arxiv.org/abs/2411.14251v3
v1 (original submission): https://arxiv.org/abs/2411.14251v1

We believe these points raise significant questions about the novelty and positioning of this submission. Proper attribution and contextualization of prior work are fundamental to scientific progress.

Therefore, we urge the authors to clarify the extensive conceptual and textual overlaps and the specific points raised here. At the same time, we respectfully ask the program committee to carefully review these substantial similarities when evaluating the paper's scientific contribution and originality.

We have provided a more detailed analysis to the ICLR Program Chairs for their review.

Sincerely,
Xidong Feng and the authors of "Natural Language Reinforcement Learning"

---

> ### Author Response · Authors · 2025-11-12
> **Author response regarding NLRL**
>
> Thank you for raising these points, and we agree that more discussion to your NLRL work and a direct comparison are definitely needed. It is clear that we had read your NLRL work and we found the idea of values in natural language space as very interesting. We do not claim to be inventing this idea and do state that we are not the first to consider RL in natural language space in Introduction and Related Work (however, we will make this point more clear in our revision). What we aim for with our work is to improve the scalability of NLRL on two fronts:
>
> (1) During policy evaluation, in order to train a natural language value function, NLRL requires aggregating on-policy samples (either full rollouts or single-step) in order to generate a natural language value. This is because values themselves are the result of an aggregation. We believe this isn’t scalable to environments with complex dynamics where it isn’t feasible to enumerate over all environment responses (such as dialogue). In our work, we aim to instead learn a natural language successor model that probabilistically generates possible futures. This can be done using temporal-difference learning on a single sample. Therefore, our language Bellman backup is to train a successor model that generates future rollouts rather than a value function.
>
> (2) During policy improvement, NLRL provides multiple candidate actions and their evaluations and performs an "argmax" over them in natural language space. In our settings, the action space is too large for such enumeration to be tractable, so we instead consider iterative refinement of actions using natural language values. Thus, it is important for our natural language values to explain why an action is suboptimal (which we use the successor model for). We believe the "refinement policy" that we propose behaves differently than language policy improvement in NLRL, and is more similar to self-refine/self-correct approaches of prior works.
>
> Regarding the statement that NLRL requires Monte-Carlo samples, we sincerely apologize as we did not notice that NLRL also works with single-step samples. However, we still believe the differences highlighted in (1) make the exact Bellman objective that we consider different than what NLRL does, as our intention was to propose a different way to learn natural language values that works even for complex environments that cannot be enumerated.  If we mischaracterized what NLRL does in any other way, we will correct those mistakes promptly. We also will make it more clear in our work that "natural language value" is not an original concept and our efforts were spent in trying to make NLRL more scalable to general LLM agent tasks.
>
> Regarding empirical comparisons, the only reason we did not initially evaluate on NLRL was because we thought adapting NLRL to our tasks would be difficult due to the scalability concerns that we had listed in (1) and (2). However, we completely agree with the authors that making this comparison is important and are currently working on how to faithfully adapt NLRL to the settings we consider.

---

> > ### Public Comment · ~Xidong_Feng1 · 2025-11-13
> >
> > Thank you for your quick and thoughtful response. We appreciate your apology regarding the mischaracterization of our work's TD learning component and your commitment to adding a direct empirical comparison and clarifying the origin of the "natural language value" concept. These are positive and important steps.
> >
> > However, we believe there remains misunderstanding in how your work is positioned relative to ours, specifically concerning your two main arguments on scalability. We would like to offer some clarification.
> >
> > **On Policy Evaluation (Distributional Successor Model vs. Language Value Function):**
> >
> > In NLRL, our goal is to learn a Language Value Function, which produces a single, coherent textual evaluation. This is analogous to a traditional value function, which represents an expectation over future outcomes. In contrast, your Language Successor Model is trained to learn a distribution over possible future rollouts. This is a fascinating direction, drawing clear inspiration from the principles of distributional RL. Perhaps a more precise framing would be a 'distributional language successor feature,' given that traditional successor features are also defined as expectations, not probabilistic distributions.
> >
> > While this is an important distinction, your "language successor model" and "language evaluator" appear to be a two-step implementation of our single-step "Language Value Function" (especially TD form). Decomposing this process is an interesting engineering choice, but the underlying function—predicting potential future outcomes (your build a future world model vs we use the sample from environment) and then aggregate to evaluate a present action—remains identical to the core mechanism of NLRL. The difference lies in learning a distribution versus an expectation. This is a valuable contribution, but it should be framed as an extension within the NLRL paradigm, not as a separate, more scalable framework built to overcome a non-existent limitation of our work.
> >
> > For the enumeration issue, you argue that NLRL is not scalable because it requires aggregating or "enumerating" environment responses, which you believe is infeasible for complex tasks like dialogue. However, we believe this challenge of handling complex, high-branching-factor environments is fundamental to any learning-based approach in this space, including your own.
> > A Language Successor Model, just like a Language Value Function, cannot learn an accurate and robust representation of an environment's dynamics if it is only ever trained on single, non-representative samples. To learn an unbiased model of possible futures, one must still sufficiently explore and sample the space of outcomes. This is, in essence, the same fundamental challenge of exploration that you claim our method faces. Relying on a single off-policy sample does not magically solve the scalability issue; it merely shifts the problem to the quality and variance of the learned model.
> >
> > **On Policy Improvement (Refinement vs. Argmax):**
> >
> > You claim that your "refinement policy" is a novel approach because NLRL is limited to an intractable "argmax" over actions.
> > Our "Language Policy Improvement" (Operator I) is a general principle: using the textual critique from the language value function to guide the policy toward a better action. An "argmax" is the simplest instantiation for discrete action spaces. Your "refinement policy" is an effective and more sophisticated instantiation of this same principle for large, generative action spaces. As such, we see it not as a competing idea, but as a valuable implementation of our proposed Language Policy Improvement for the LLM agent setting.
> >
> > **Summary**
> >
> > In summary, the scalability solutions you present seem to fall squarely within the general framework we introduced in NLRL. We are glad that you have found our work interesting and are exploring these implementations. Our primary concern has been that these implementation details were presented as fundamental innovations that set your work apart. As you can tell, the entire conceptual architecture of your paper—from the language-based critic to the off-policy updates and the language-guided policy improvement—mirrors the framework we had already proposed and discussed in detail. This perhaps explains why the overall structure of your paper is so similar to ours: the foundational blueprint was already laid out. Acknowledging this is a key point that is critically missing from your paper's current narrative.
> >
> > We are encouraged by your commitment to revise the paper. We hope the updated version will accurately frame NLAC not as a new framework, but as a specific and valuable implementation and extension of the NLRL framework, adapted with smart engineering choices for complex agentic tasks and scalable training. We believe this would clarify your true contributions while giving proper credit to the foundational concepts.
> >
> > We look forward to seeing the revision and the empirical comparison.

---

> > > ### Author Response · Authors · 2025-11-13
> > > **Author Response**
> > >
> > > Thank you for your response. We understand the points that you are raising above, and will adjust the narrative of our work as contributing on top of the paradigm of algorithms introduced by NLRL.

---

### Author Response · Authors · 2025-11-21
**Joint Response Regarding Revisions to Paper**

We would like to thank all the reviewers for their helpful reviews. After aggregating feedback from all the reviews, we made several key revisions to the paper:

**Theoretical Analysis of NLAC**

We added a section on theoretical analysis showing our method can converge to the optimal policy. Traditional analyses do so by showing convergence of learned Q-values to the ground-truth, then monotonic improvement of the policy by optimizing such Q-values. However, in our setting, we never explicitly learn Q-values that are scalars. The key insight we leverage to circumvent this difficulty is that when learning our successor function, we show that the underlying representations converge to the unique successor features of the MDP [1]. This gives us a way to connect our learned language values to the true scalar Q-values.

**Improvements to algorithm description**

Multiple reviewers raised some important concerns about the effectiveness of our method, namely that multiple components of our method are not explicitly trained, but rely on the base reasoning capabilities of the LLM via careful prompting. These are concerns that we also had during the development of our method.

We add two new subsections explaining modifications that we considered making to our algorithm to address these concerns, namely: (1) explicitly training the refinement policy using policy optimization on achieved reward after refinement, and (2) regularization to the initial LLM model to mitigate catastrophic forgetting. Overall, we excluded these modifications from the default implementation of our method because we found that though they improved stability (allowing for training for more iterations), they ultimately did not improve final performance by a significant enough margin.

We also added figures to our revised paper showing how the different components of our method interact, as we realized that the description on its own was dense and potentially difficult to parse.

**Comparison to NLRL**

The most similar prior method to ours is NLRL [2], which (to our knowledge) is the first work to propose learning a critic for RL fine-tuning that outputs evaluations in natural language space. We discuss our contribution on top of NLRL in more detail in the revised paper, but on a high-level, our algorithm NLAC falls under the paradigm of algorithms introduced by NLRL, with the goal of being more scalable and generalizable to complex tasks. Specifically, our algorithm NLAC aims to improve upon two aspects of the NLRL that we believed made it difficult to scale to general LLM agent tasks:

(1)  During policy evaluation, the critic in NLRL is trained by aggregating multiple on-policy samples in-context and generating a holistic evaluation using these samples. Such on-policy samples can simply consist of next state with future predicted value (like in standard temporal difference learning), but for complex tasks, we found enumerating over environment transitions in-context for one training sample to be too expensive. In NLAC, rather than training the critic to output aggregated evaluations, we train a successor model to probabilistically generate rollouts. We found that such training can be done using a single transition like in traditional RL, is theoretically motivated by learning successor features, and can later be used for action refinement.

(2) During policy improvement, NLRL provides multiple candidate actions and their evaluations and performs an "argmax" over them in natural language space. In our settings, the action space is too large for such enumeration to be tractable, and we found empirically that simply sampling a tractable subset of actions does not usually adequately explore the action space. Hence, in NLAC, we consider iterative refinement of the policy. In order to refine the policy, the evaluation must explain how an action can be improved, which we provide by including a description of what will likely happen afterwards.

Because of those drawbacks, we initially believed NLRL would be intractable to implement for our experiments, but we have hence decided to evaluate against it, only limiting the number of samples that are fit in-context. Thus, in the revised paper, we compare against NLRL, where we limit to 8 in-context transitions and actions to keep training tractable; note that this can be interpreted as providing 8 times more samples during training. NLRL performs competitively in both math reasoning and 20Q, but falls short in \tau-bench. We found that in \tau-bench, which has the most complex dynamics and action space, the language values learned by NLRL were always positive in sentiment. We believe this could be due to not seeing enough in-context samples during training to overcome the implicit positive bias of instruction-fine-tuned LLMs.

[1] https://arxiv.org/abs/1606.05312

[2] https://arxiv.org/abs/2411.14251

---

### Meta-Review · Area_Chair_6G7z · 2025-12-18

**Summary:**

This paper received initial reviews which were borderline (4, 4, 4, 6). Separately from this, concerns were raised in an external comment by the authors of a closely related prior paper [1]. These concerns claimed that prior work [1] was misrepresented, and key concepts introduced in [1] were rebranded by the present paper as novel contributions. Following this, there was some discussion between the authors of this paper and the authors of [1] around the similarities and differences of the two papers. This discussion ended without a clear resolution, with the authors promising to update their paper, but leaving several of the points raised by the authors of [1] unanswered.

I have read both papers in detail and will first discuss the concerns raised by the external commenter, and then the discussion with the reviewers in the next section. Overall, I think the paper is unfit for publication at ICLR both due to incorrectly representing closely related prior work [1] (see below), and because it does not meet the bar for scientific rigor (see next section), hence I am recommending **reject**.

Unfortunately, the updated paper did not mark its changes in a different color, and the original submission is unavailable, therefore it is difficult to know what exact changes were made during the rebuttal process. Therefore, my assessment is based on the current version of the paper.

- The present paper still introduces several of the concepts introduced by [1] under different names, which carries the implication that they are novel. I think this paper should keep the names used in [1] for the common components, and describe them in a background section rather than their methods section.
- The present paper still claims that [1] is not scalable because it requires enumeration over all actions/environment transitions and fitting them in context when estimating the value. This enumeration does not appear anywhere in [1], which instead approximates language value functions through sampling, as is done in classic RL and in this paper. The authors of [1] had clarified this point in their comment.
- The present paper still claims that [1] is limited to performing argmax operations over actions in its policy improvement step. I have checked the paper and the policy improvement step is described as a generic LLM prompt asking to improve the current action, and is not necessarily limited to an argmax operation.

It is also telling that 3/4 reviewers understood this paper as introducing the idea of learning natural language value functions, which was in fact introduced by [1]:
- Reviewer mvfc: “They propose Natural Language Actor-Critic (NLAC), which replaces a scalar critic with a generative, natural-language critic..”
- Reviewer 6UAb: “This paper proposes Natural Language Actor-Critic (NLAC), a new actor-critic algorithm designed to train LLM agents […] NLAC replaces the traditional scalar-value critic with a generative LLM critic that produces textual critiques of actions.”
- Reviewer MUW2: “The novelty lies mainly in the use of a generative critic, that produces feedback predictions on the future of trajectories, rather than -- less-informative -- numerical rewards”



[1] "Natural Language Reinforcement Learning",  X. Feng, B Liu, et al.  (https://arxiv.org/abs/2411.14251).

**Reviewer Concerns:**

Separately from the above, there are several issues (some brought up by the reviewers), which would need to be addressed before the paper is ready for publication:

- Error bars are not included in the main table (Table 1), making it unclear which differences are statistically significant or not (this was brought up by reviewer JPr6). Figure 2 does have error bars, but the caption does not say what the represent (standard deviation, 1 or 2 standard errors, confidence intervals?). Also, Figure 2 does not say what task the curves are for.
- Given the very close similarities between the proposed work and NLRL [1], a more fine-grained comparison than the one provided is necessary. The two differences are: a different version of the policy improvement operator, and a different implementation of the natural language critic update. A cross product ablation of the 4 different combinations would clarify where the performance differences are coming from (currently this is not clear - this was also suggested by reviewer 6UAb). Furthermore, there aren't many experimental details of the NLRL implementation in the Appendix (e.g. was the original codebase used? Did you re-implement it? Did both methods include the same design components, such as prioritized experience replay, EMA, etc?). Adding these is especially important given the similarities between the two works.
- The newly added theoretical analysis makes a strong assumption (5.2), that is not standard in the literature and which they do not give any evidence holds in practice. Related to this, reviewer 6UAb asked what would happen if the LLM recommends suboptimal actions during the policy improvement step (which could very well happen). I would think any theoretical results should include some term which measures how well the LLM improves the policy. The current result says that the described procedure will converge to the optimal policy, which is clearly not the case based on experiments – therefore, the assumption must break somewhere in practice, but it is not clear where. It is unclear what the theoretical result adds to our understanding of the algorithm.
- Reviewer mvfc asked for several ablations, which I agree would be helpful (number of refinement rounds m, number of possible futures sampled k, effect of prioritized experience replay, etc. However, none of these were provided.

**Reviewer Scores:**

mvfc: 4 -> 4

6UAb: 4 -> 4

MUW2: 6 -> 6

JPr6: 4 -> 6

---

### Decision · Program_Chairs · 2026-01-26

Reject